# Activity in developing prefrontal cortex is shaped by sleep and sensory experience

Lex J Gómez[1]*, James C Dooley[2,3], Mark S Blumberg[1,2,3,4]*

[1]Interdisciplinary Graduate Program in Neuroscience, University of Iowa, Iowa City, United States; [2]Department of Psychological and Brain Sciences, University of Iowa, Iowa City, United States; [3]DeLTA Center, University of Iowa, Iowa City, United States; [4]Iowa Neuroscience Institute, University of Iowa, Iowa City, United States

**Abstract** In developing rats, behavioral state exerts a profound modulatory influence on neural activity throughout the sensorimotor system, including primary motor cortex (M1). We hypothesized that similar state-dependent modulation occurs in prefrontal cortical areas with which M1 forms functional connections. Here, using 8- and 12-day-old rats cycling freely between sleep and wake, we record neural activity in M1, secondary motor cortex (M2), and medial prefrontal cortex (mPFC). At both ages in all three areas, neural activity increased during active sleep (AS) compared with wake. Also, regardless of behavioral state, neural activity in all three areas increased during periods when limbs were moving. The movement-related activity in M2 and mPFC, like that in M1, is driven by sensory feedback. Our results, which diverge from those of previous studies using anesthetized pups, demonstrate that AS-dependent modulation and sensory responsivity extend to prefrontal cortex. These findings expand the range of possible factors shaping the activity-dependent development of higher-order cortical areas.

## Editor's evaluation

This manuscript examines neural activity in several cortical areas (such as the primary and secondary motor cortex and the medial prefrontal cortex) across sleep-wake states and under anesthesia. The quality of the recordings in infant rats is excellent, evidence is solid, and results are important in the field of research into the role of active sleep in the neuronal and circuit mechanisms of early cortical development. Some of the findings presented and the hypothesis developed are novel, and some should hopefully prompt future developmental studies to look at sleep as an essential component that cannot be replaced by using anesthetics.

*For correspondence:
lexcience@gmail.com (LJG);
mark-blumberg@uiowa.edu
(MSB)

**Competing interest:** The authors declare that no competing interests exist.

## Introduction

The functional development of cerebral cortex is a sinuous and often surprising process, even for those structures with the most transparent of adult functions. Consider primary motor cortex (M1), whose name reflects its well-established role in adult motor control: In early development, M1 does not contribute to motor control at all, but instead functions exclusively as a sensory structure (**Bruce et al., 1980**; **Chakrabarty and Martin, 2005**; **Young et al., 2012**; **Tiriac et al., 2014**; **Dooley and Blumberg, 2018**; **Singleton et al., 2021**). The early-emerging somatosensory map in M1 provides the foundation upon which its later-emerging motor map is built (**Dooley and Blumberg, 2018**).

Another surprising aspect of M1 in early development is that its activity is modulated by behavioral state, in particular active sleep (AS, or REM sleep). In infant rats, this modulation reflects AS-dependent increases in neural activity that are enhanced by limb movements during AS, called twitches, that discretely and preferentially trigger sensory feedback to M1 (**Dooley and Blumberg, 2018**; **Glanz**

*et al., 2021*). Importantly, AS-dependent modulation of activity is not unique to M1 but is seen in many developing sensorimotor structures (*Blumberg, 2015*; *Blumberg et al., 2020*). Given that AS predominates in early life (*Jouvet-Mounier et al., 1969*; *Gramsbergen et al., 1970*), it has been posited that this sleep state plays an outsized role in typical and atypical development (*Blumberg et al., 2022*).

That sleep so profoundly modulates neural activity in developing sensorimotor structures raises the possibility that it also modulates activity in cortical areas that are directly or indirectly influenced by sensorimotor input, including higher-order areas like prefrontal cortex. Of particular interest here are two areas with which M1 forms connections: secondary motor cortex (M2) and medial prefrontal cortex (mPFC) (*Van Eden et al., 1992*; *Bedwell et al., 2014*; *Bedwell et al., 2017*). As its name implies, M2 has a particularly close functional and anatomical connection with M1: It integrates multi-modal sensory cues for motor planning and modulates M1 activity during goal-directed action (*Yin, 2009*; *Omlor et al., 2019*; *Barthas and Kwan, 2017*; *Morandell and Huber, 2017*; *Wang et al., 2020*). Like M1, M2 develops a somatotopic map, further highlighting its dependence on sensory input (*Yin, 2009*; *Kunori and Takashima, 2016*; *Omlor et al., 2019*; *Barthas and Kwan, 2017*; *Chen et al., 2017*; *Singleton et al., 2021*). Thus, we hypothesize that, in infant rats, M2 is similar to M1 with respect to sensory responsiveness and modulation by behavioral state.

In contrast with M1 and M2, mPFC in adults is not closely associated with sensorimotor functions, but rather with cognitive processes such as decision-making and attention (*Tanji and Hoshi, 2001*; *Tanji and Hoshi, 2008*; *Miller et al., 2002*; *Barbas and Zikopoulos, 2007*; *Euston et al., 2012*). In infant rats, it is not known whether behavioral state modulates activity in mPFC, nor is it known whether mPFC processes sensory input. In fact, it has been theorized that prefrontal cortex, including mPFC, develops its unique higher-order functions precisely because it develops independently of sensory input (*Johnson et al., 2015*).

What is currently known about functional development in mPFC derives primarily from neural recordings from rat pups under urethane anesthesia (*Brockmann et al., 2011*; *Bitzenhofer et al., 2015*). Although urethane precludes natural sleep–wake cycles, it does not prevent expression of spindle bursts in mPFC (*Brockmann et al., 2011*). Spindle bursts are brief thalamocortical oscillations that, in primary sensory areas, are closely associated with the processing of sensory stimuli (*Khazipov et al., 2004*; *Hanganu et al., 2007*; *Dooley et al., 2020*). In the mPFC of urethanized pups, however, spindle bursts appear to occur spontaneously. Here, we determine if this is also the case in unanesthetized pups—as well as test the hypothesis that the infant mPFC, like M1, is modulated by behavioral state.

Using unanesthetized rats at postnatal days (P) 8 and P12, we find that M2 and mPFC exhibit state-dependent modulation such that neural firing rates are highest during AS, especially during periods of twitching. We also find that neurons in M2 and mPFC respond to sensory input arising from limb movements that are self-generated (i.e., reafference) or other-generated (i.e., exafference). Finally, to explain discrepancies between the present findings and those reported earlier, we show that urethane administration at P8 prevents expression of behavioral state and brain–behavior relations. Altogether, these findings demonstrate that previously documented effects of behavioral state and sensory experience on somatosensory activity in M1 extend to M2 and mPFC, thus pointing toward new directions for conceptualizing activity-dependent development of higher-order cortical areas.

## Results

We recorded extracellular unit activity in M1, M2, and mPFC in head-fixed rats at P8 and P12 (*Figure 1A*). For each pup, dual recordings were performed first in the forelimb regions of M1 and M2 (also referred to as the caudal and rostral forelimb areas, respectively) for 40 min, followed by 50 manual stimulations of the forelimb contralateral to the recording sites. Next, the M2 electrode was repositioned in mPFC and the recording and stimulation procedure was repeated but now with dual recordings in M1 and mPFC. Electrode locations in M1, M2, and mPFC were confirmed histologically (*Figure 1B*). At P8, we collected eight M1–M2 recordings (107 M1 units, 118 M2 units) and eight M1–mPFC recordings (117 M1 units; 103 mPFC units); at P12 we collected nine M1–M2 recordings (217 M1 units; 204 M2 units) and eight M1–mPFC recordings (222 M1 units; 179 mPFC units). Neural activity, electromyographic (EMG) activity in the nuchal and biceps muscles, and high-speed video (100 frames/s) were recorded as pups cycled between sleep and wake (*Figure 1C, D*). As expected

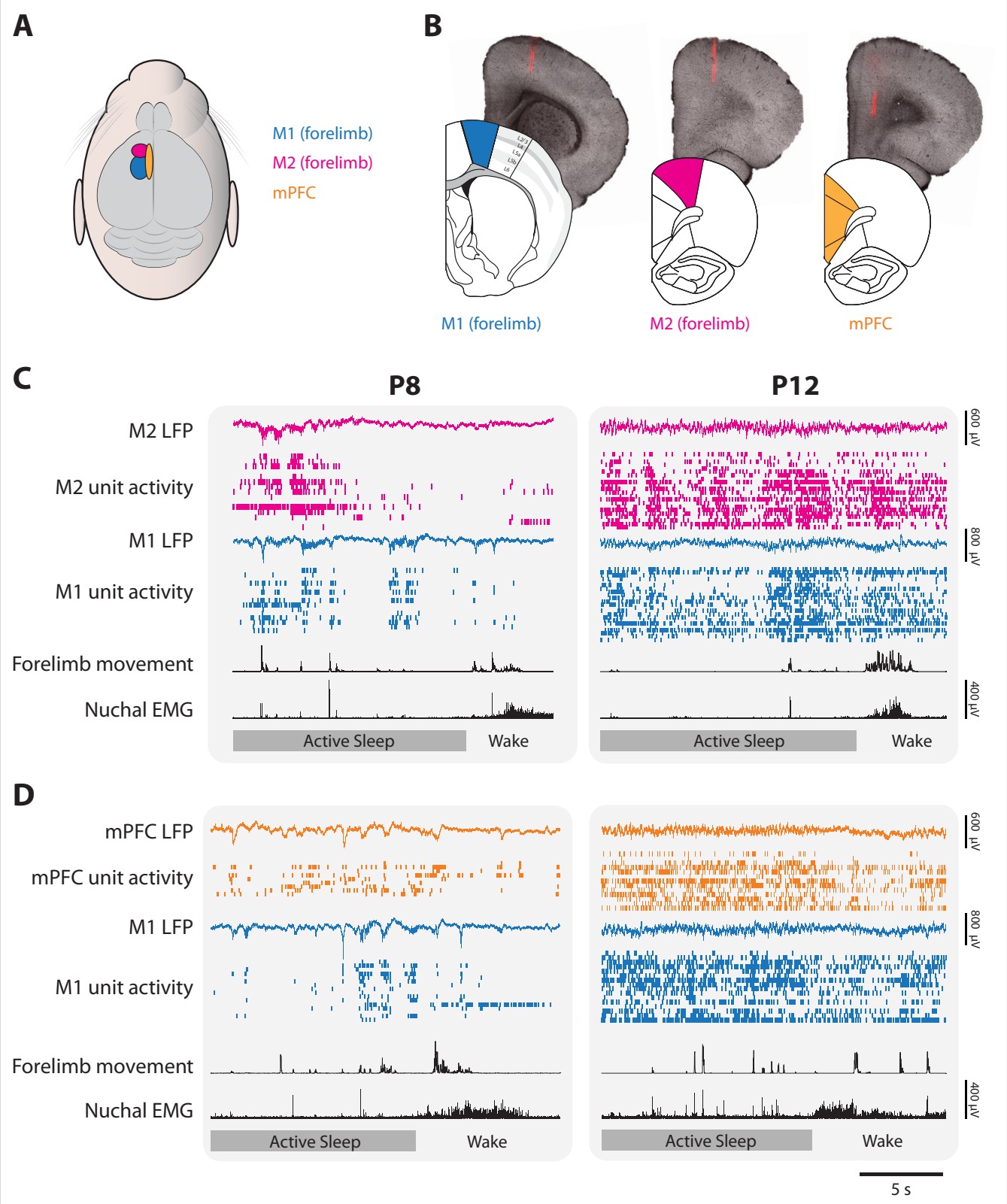

**Figure 1.** Representative neural activity in M1, M2, and mPFC in P8 and P12 rats. (**A**) Illustration showing the surface locations of M1 (blue), M2 (magenta), and mPFC (gold). These color codes are used in all other figures. (**B**) From left to right, illustrations of coronal sections of M1, M2, and mPFC beneath corresponding brightfield coronal sections that show a fluorescent electrode track in each area. (**C**) Representative 20-s segments of data from paired recordings in M1 and M2 at P8 (left) and P12 (right) across behavioral states. For each record from the top, data are presented as follows: M2

*Figure 1 continued on next page*

Figure 1 continued

local field potential (LFP) (magenta trace), M2 unit activity (magenta ticks), M1 LFP (blue trace), M1 unit activity (blue ticks), forelimb movement, and nuchal electromyography (EMG). Bottom row: Behavioral states marked as active sleep (dark gray) or wake (light gray). (D) Same as in C, but for paired recordings in M1 and mPFC (mPFC LFP, gold trace; mPFC unit activity, gold ticks).

(*Dooley et al., 2021*; *Glanz et al., 2021*; *Gómez et al., 2021*), pups spent more time in AS than wake at P8 (AS: 57.7 ± 2.5%; wake: 30.9 ± 1.9%) and P12 (AS: 44.0 ± 3.6%; wake: 39.3 ± 3.5%). Also, the transition from discontinuous cortical activity at P8 to continuous activity at P12 was evident in all three areas, as described previously in primary somatosensory, motor, and visual cortex (*Golshani et al., 2009*; *Rochefort et al., 2009*; *van der Bourg et al., 2017*; *Glanz et al., 2021*; *Riyahi et al., 2021*).

## Neural activity in M2 and mPFC is modulated by behavioral state

At P8, representative recordings in M1, M2, and mPFC illustrate substantial and often abrupt increases in neural activity during AS (*Figure 2A*). In each area, the mean firing rate was significantly higher during AS than wake ($t_{(7)}$s ≥ 4.38, ps ≤ 0.003, Cohen's $D$s ≥ 1.55; *Figure 2B*; see *Figure 2—figure supplement 1* for data from representative recordings). State-dependent modulation of cortical activity continued through P12 (*Figure 2C*); once again, the mean firing rate in each area was significantly higher during AS than wake ($t_{(7–8)}$s ≥ 3.17, ps ≤ 0.016, Cohen's $D$s ≥ 1.12, *Figure 2D*). Similarly, at P8, the rate of spindle bursts was higher during AS than wake for all three areas ($t_{(7)}$s ≥ 3.805, ps ≤ 0.007, Cohen's $D$s ≥ 1.35; *Figure 3*) and were associated with increases in unit activity (*Figure 3—figure supplement 1*). (Spindle bursts were not analyzed at P12 as they are not clearly discernable at this age.) Thus, like M1, neural activity in M2 and mPFC is modulated at both ages in a state-dependent manner.

## Neural activity in M2 and mPFC increases during periods of self-generated movement

In infant rats, AS-dependent increases in M1 activity correspond with periods of limb movement (e.g., *Glanz et al., 2021*). Thus, we next determined whether the same is true for M2 and mPFC (*Figure 4A*). At both ages, the mean firing rate in each area increased significantly during periods of movement (*Figure 4B*). For all cases, repeated-measures analyses of variance (ANOVAs) revealed significant main effects of behavioral state ($F_{(1,7–8)}$s ≥ 16.90, ps ≤ 0.005, $\eta_p^2$s ≥ 0.71) and movement ($F_{(1,7–8)}$s ≥ 6.83, ps ≤ 0.035, $\eta_p^2$s ≥ 0.49). None of the state × movement interactions was significant ($F_{(1,7–8)}$s ≤ 5.35), except for one of the M1 tests at P8 ($F_{(1,7)}$ = 6.11, p = 0.043, $\eta_p^2$ = 0.47).

For each of the eight repeated-measures ANOVAs across P8 and P12, four planned comparisons were conducted to compare firing rates within behavioral state across movement conditions and within movement conditions across behavioral state (*Figure 4B, C*). Of the 32 planned comparisons, 31 were significant ($t_{(7–8)}$s ≥ 3.51, ps ≤ 0.01, Cohen's $D$s ≥ 1.24). The general pattern was for firing rates to be highest during AS-related periods of movement (i.e., twitching), intermediate during periods of AS-related periods of no movement and wake-related periods of movement, and lowest during wake-related periods of no movement.

In summary, at P8 and P12, neural activity in M1, M2, and mPFC reflects the interactive effects of behavioral state and movement. Given that all three areas exhibited similar movement-related increases in activity and that M1 is known to respond to movement-related sensory feedback (*Dooley and Blumberg, 2018*; *Gómez et al., 2021*), we determined next whether M2 and mPFC are also responsive to sensory input.

## Neurons in M2 and mPFC respond to sensory input

We quantified neural responses in M1, M2, and mPFC to forelimb twitches, wake movements, and stimulations (*Figure 5A*). As expected (and in M1, consistent with previous results; *Tiriac et al., 2014*; *Dooley and Blumberg, 2018*; *Glanz et al., 2021*; *Gómez et al., 2021*), units in M1 and M2 at both ages responded to sensory input arising from twitches, wake movements, and stimulations; surprisingly, so did units in mPFC. The percentage of responsive units in all three areas varied by age and event type. At P8, M1 generally exhibited the highest percentage of responsive units, followed by M2 and then mPFC ($t_{(7)}$s ≥ 4.44, ps ≤ 0.003, Cohen's $D$s ≥ 1.57; *Figure 5B*). Mean M2 responsiveness was 60.9 ± 10.2% for twitches, 46.5 ± 11.9% for wake movements, and 36.7 ± 9.1% for stimulations; for

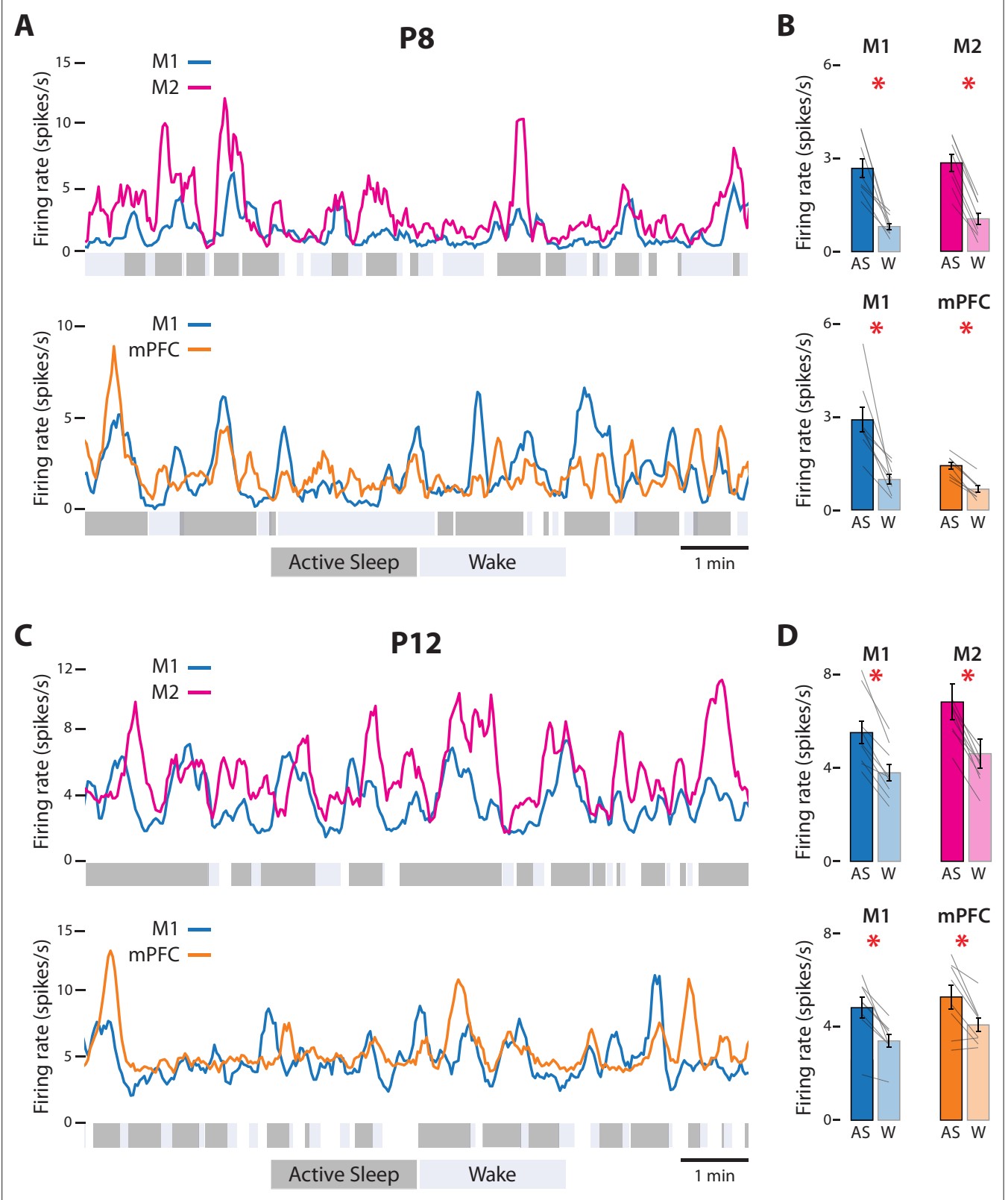

**Figure 2.** State-dependent unit activity in M1, M2, and mPFC in P8 and P12 rats. (**A**) Representative 10-min segments of data from a P8 rat showing mean firing rate (2-s bins) in relation to active sleep (dark gray) and wake (light gray). Top: Units in M1 and M2. Bottom: Units in M1 and mPFC. (**B**) Top: Mean firing rates for M1 and M2 units during active sleep (AS) and wake (W). Bottom: Mean firing rates for M1 and mPFC units during AS and wake. Mean firing rates for individual pups are shown as gray lines. Means ± standard error of the mean (SEM). Asterisks denote significant difference between

*Figure 2 continued on next page*

*Figure 2 continued*

states, p ≤ 0.025. (**C**) Same as in A, but for a P12 rat. (**D**) Same as in B, but for P12 rats. (For M2, the values for one data pair exceed 8 spikes/s and are not shown.)

The online version of this article includes the following figure supplement(s) for figure 2:

**Figure supplement 1.** State-dependent activity of individual units in M1, M2, and mPFC.

mPFC, these values were 37.8 ± 8.0%, 20.6 ± 7.5%, and 9.4 ± 5.5%, respectively. At P12, responsiveness declined to low levels in all three areas, but M1 was still more responsive than M2 or mPFC ($t_{(7–8)}$s ≥ 3.13, ps ≤ 0.017, Cohen's $D$s ≥ 1.11; *Figure 5C*).

Regardless of the mean responsiveness of a cortical area at a given age, when units were responsive they exhibited response profiles (i.e., perievent time histograms, PETHs) that were strikingly similar to each other. These profiles indicate sensory responding because the peaks in activity occurred after the

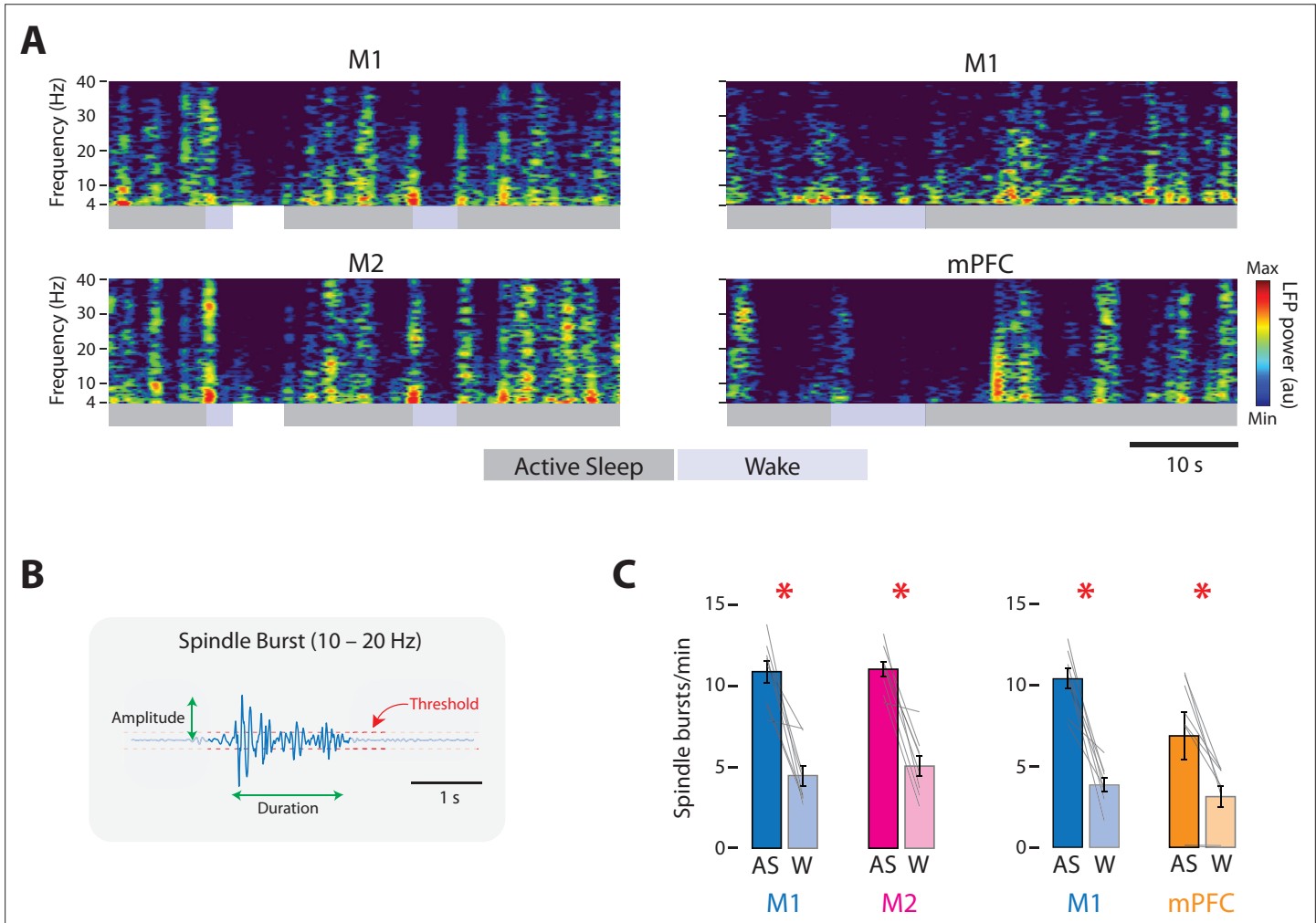

**Figure 3.** State-dependent spindle-burst activity in M1, M2, and mPFC in P8 rats. (**A**) Left column: Representative 50-s segment of local field potential (LFP) data showing spindle bursts in the spectrogram for a paired M1 (top) and M2 (bottom) recording across active sleep (dark gray) and wake (light gray). Right column: Same as for left column, but for a paired M1 (top) and mPFC (bottom) recording. (**B**) Illustration to show method for detecting spindle bursts and calculating their amplitude and duration. Spindle bursts were defined when the median LFP amplitude exceeded, for at least 100 ms, an established threshold (horizontal dashed lines). (**C**) Bar graphs showing mean spindle-burst rate in M1, M2, and mPFC during active sleep (AS) and wake (W). Mean firing rates for individual pups are shown as gray lines. Means ± standard error of the mean (SEM). Asterisks denote significant difference between states, p ≤ 0.025.

The online version of this article includes the following figure supplement(s) for figure 3:

**Figure supplement 1.** Unit activity associated with spindle bursts in M1, M2, and mPFC.

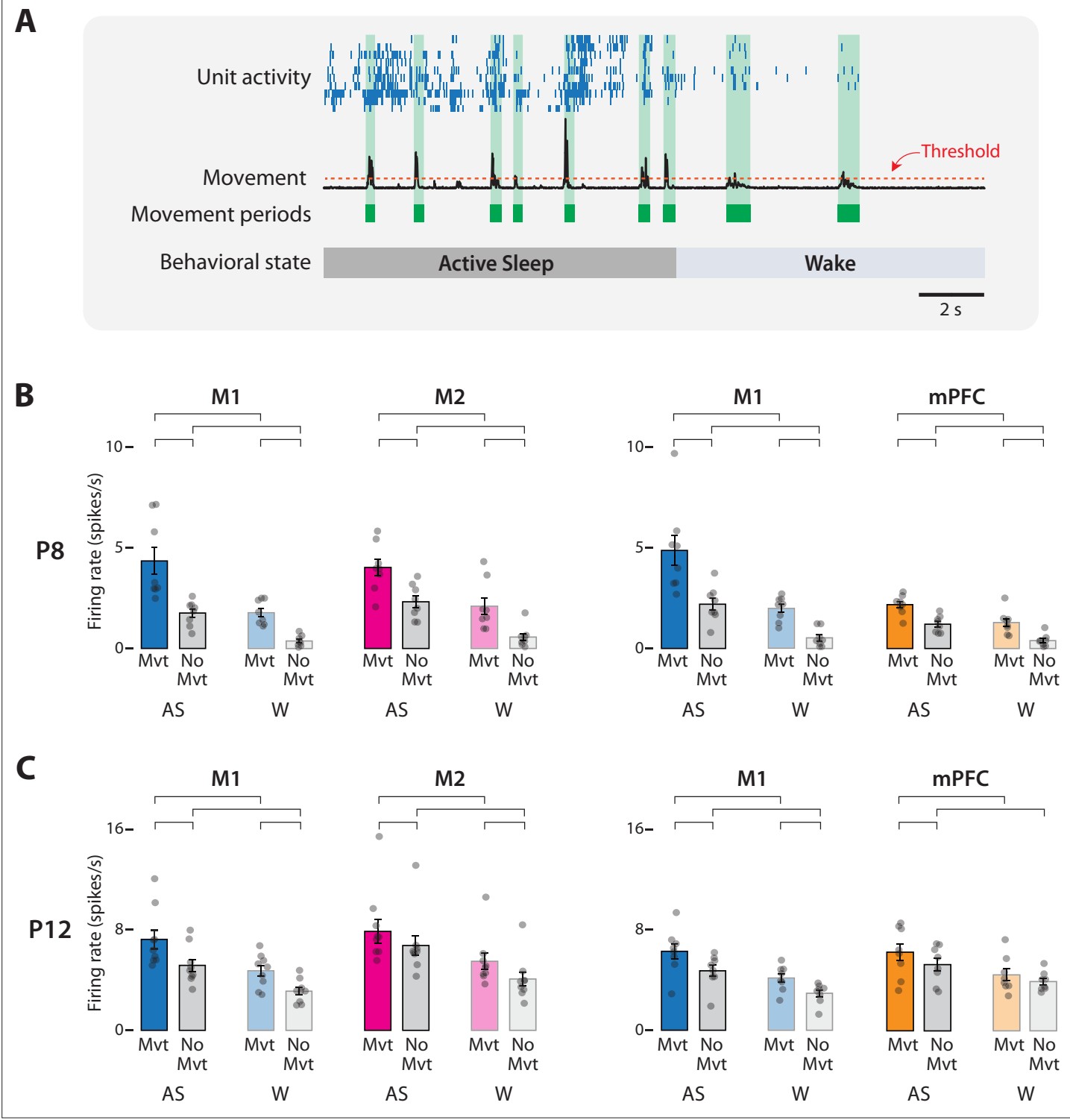

**Figure 4.** Movement-dependent unit activity in M1, M2, and mPFC in P8 and P12 rats. (**A**) Representative 20-s segment of data showing unit activity (blue ticks), movement data (black trace), movement periods (green blocks), movement-detection threshold (horizontal dotted line), and behavioral state. (**B**) Bar graphs showing mean firing rates for neurons in M1, M2, and mPFC during periods of movement (Mvt) or no movement (No Mvt) across active sleep (AS) and wake (W). Mean firing rates for individual pups are shown as gray circles. Means ± standard error of the mean (SEM). Brackets denote significant difference between groups, p ≤ 0.0125. (**C**) Same as in B, but for P12.

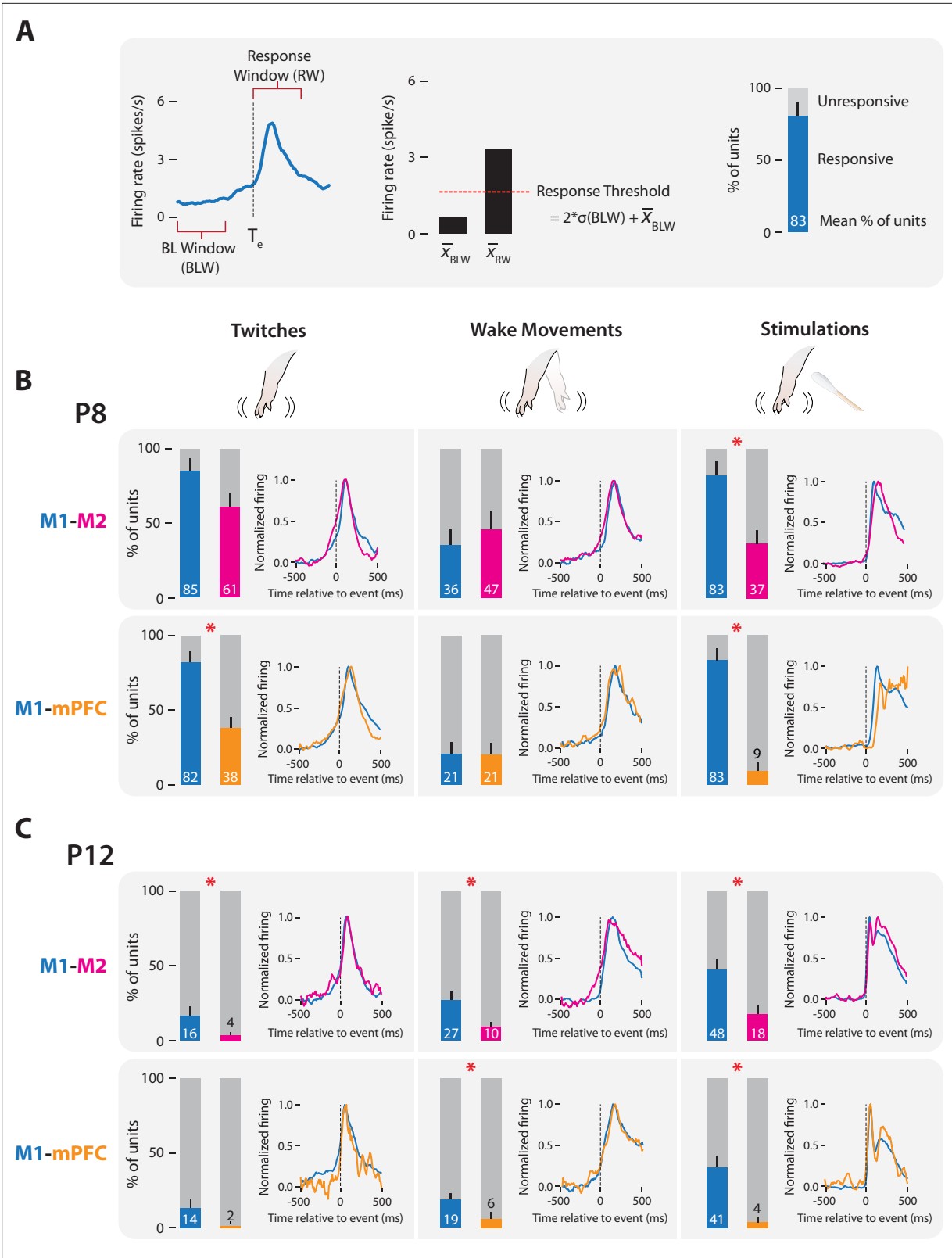

**Figure 5.** M1, M2, and mPFC neural responses to sensory input in P8 and P12 rats. (**A**) Methodology for determining sensory responsiveness of individual units. Left: Perievent time histogram (PETH) of unit firing rate (blue line) relative to a sensory event, showing the baseline window (BLW) and response window (RW). Event onset denoted by dotted line at $T_e$. Middle: Bar graph showing mean unit activity during the BLW and RW, response threshold (dotted line), and the threshold calculation. Right: Stacked plot showing the percentage of units that exceeded the response threshold (blue)

*Figure 5 continued on next page*

*Figure 5 continued*

and percentage of units that did not (gray). (**B**) Stacked plots showing mean percentage of responsive (colored) and unresponsive (gray) units across pups at P8. Top: Data from M1 (blue) and M2 (magenta) recordings. Bottom: Data from M1 (blue) and mPFC (gold) recordings. Means + standard error of the mean (SEM). Asterisks denote significant difference between cortical areas, p ≤ 0.017. PETHs to the right of each stacked plot show normalized unit firing rates for responsive units only in each cortical area. (**C**) Same as in B, but at P12.

movement or stimulation (*Figure 5B, C*). Such response profiles were observed even when responsive units were rare (e.g., mPFC at P12).

To assess whether M1, M2, and mPFC were similarly activated by sensory events, we next measured each area's activation rate to compare the reliability with which each area responded to sensory events. The activation rate was defined as the percentage of twitches, wake movements, or stimulations for which at least 30% of responsive units in the area showed an increase in activity (*Figure 6A*); this threshold was chosen based on prior study of population responses in M1 to self-generated movements at P8 and P12 (*Glanz et al., 2021*). At P8, M2 and mPFC showed mean activation rates that were similar to those in M1 ($t_{(7)}$s ≤ 2.28) (*Figure 6B*), with one exception: mPFC had a significantly lower mean activation rate than M1 for stimulations ($t_{(7)}$ = 5.91, p < 0.001, Cohen's D = 2.09). At P12, mean activation rates in M2 and M1 were also similar ($t_{(8)}$s ≤ 2.53) (*Figure 6C*); for mPFC, the mean activation rate for twitches was similar to that for M1 ($t_{(7)}$ = 1.73), but mean activation rates were significantly lower in mPFC for wake movements and stimulations ($t_{(7)}$s ≥ 4.06, ps ≤ 0.005, Cohen's Ds ≥ 1.43). Thus, whereas M1 and M2 had comparable activation rates to all three kinds of sensory events at both ages, mPFC was generally less responsive than M1.

Spindle bursts in all three areas were associated with sensory events. In M1 and M2, spindle bursts were significantly more likely to occur following twitches, wake movements, and stimulations when compared to shuffled data ($t_{(7)}$s ≥ 5.79; ps ≤ 0.001; Cohen's Ds ≥ 0.93) (*Figure 6—figure supplement 1*). Likewise, spindle bursts in mPFC reliably followed wake movements and stimulations ($t_{(7)}$s ≥ 4.27; ps ≤ 0.004; Cohen's Ds ≥ 1.00), though not twitches ($t_{(7)}$ = 2.16). In most cases, sensory events were equally likely to trigger spindle bursts across areas, with the exception of twitches for M2 and stimulations for mPFC ($t_{(7)}$s ≥ 3.51, ps ≤ 0.01, Cohen's Ds ≥ 1.24).

Finally, the fact that M1, M2, and mPFC all exhibited sensory responses at these ages led us to consider the sources of this sensory input, as done previously for M1 and primary somatosensory cortex (S1; *Gómez et al., 2021*). Specifically, we performed analyses to determine if sensory input is conveyed in parallel to these structures or serially between them (e.g., from M1 to M2). However, unlike with M1 and S1, this analysis yielded a null result: Individual sensory events did not reliably trigger contemporaneous unit activity in M1 and M2, or M1 and mPFC above chance (data not shown). This result suggests that the pathways through which sensory input reaches M2 and mPFC are distinct from those that reach M1 and S1, and presently remain unknown.

In summary, as found previously in M1, both M2 and mPFC respond to sensory input in early development with increases in spiking activity and spindle bursts.

## Urethane anesthesia suppresses behavior and neural activity in M1 and mPFC

The prefrontal activity described thus far does not resemble that reported previously in urethanized pups (*Brockmann et al., 2011*; *Bitzenhofer et al., 2015*; *Chini et al., 2019*). To determine whether the use of urethane accounts for this disparity, we recorded M1 and mPFC activity in an additional set of P8 rats (*n* = 6/group) before and after administration of urethane (1.0 mg/g b.w. SC) or sterile saline.

Urethane administration produced rapid and dramatic effects on behavior and neural activity (*Figure 7A*). Before urethane injection, pups cycled between sleep and wake, as evidenced by alternating periods of high and low muscle tone accompanied by bouts of wake movements and twitches, respectively. In contrast, urethane (but not saline) injection produced muscle atonia (with occasional spasmodic increases in muscle tone) and suppressed limb movements, thus precluding identification of sleep–wake states. Compared with saline, urethane injection produced significant percentage decreases in limb movements (urethane: −87.14 ± 6.75%; $t_{(5)}$ = 11.90, p < 0.001, Cohen's D = 4.86; saline: +10.84 ± 5.84%, $t_{(5)}$ = 1.00). Of the limb movements that remained after urethane, most occurred during brief whole-body spasms; twitch-like movements were rarely observed.

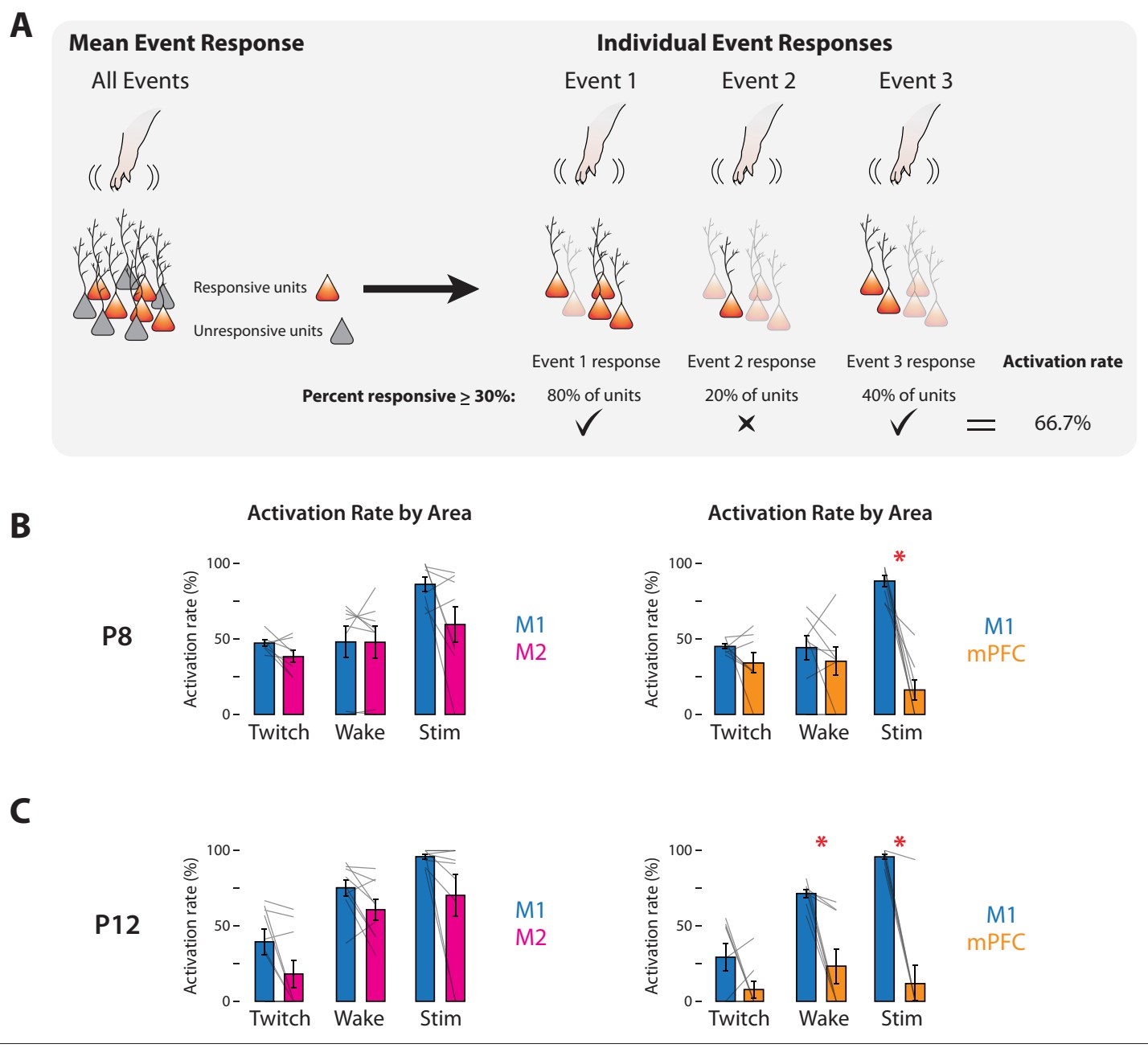

**Figure 6.** Response rates of M1, M2, and mPFC to sensory events in P8 and P12 rats. (**A**) Methodology for determining the activation rate of cortical areas to sensory events. Left: Illustration of responsive (orange) and unresponsive (gray) units within an area. Right: Illustration of activity of responsive neurons (opaque) to individual sensory events. For each of the three sensory events indicated, the percentage of responsive units is determined. Based on the percentage of events that exceeds threshold (≥30%; check marks), the activation rate is calculated. (**B**) Activation rates in M1, M2, and mPFC at P8 to twitches (Twitch), wake movements (Wake), and stimulations (Stim). Left: Bar graphs showing percentage of sensory events that evoked a response in M1 and M2. Right: Same as at left, but for M1 and mPFC. Mean activation rates for individual pups are shown as gray lines. Means ± standard error of the mean (SEM). Asterisks denote significant difference between areas, p ≤ 0.017. (**C**) Same as in B, but at P12.

The online version of this article includes the following figure supplement(s) for figure 6:

**Figure supplement 1.** Spindle bursts rates in M1, M2, and mPFC in response to sensory events in P8 rats.

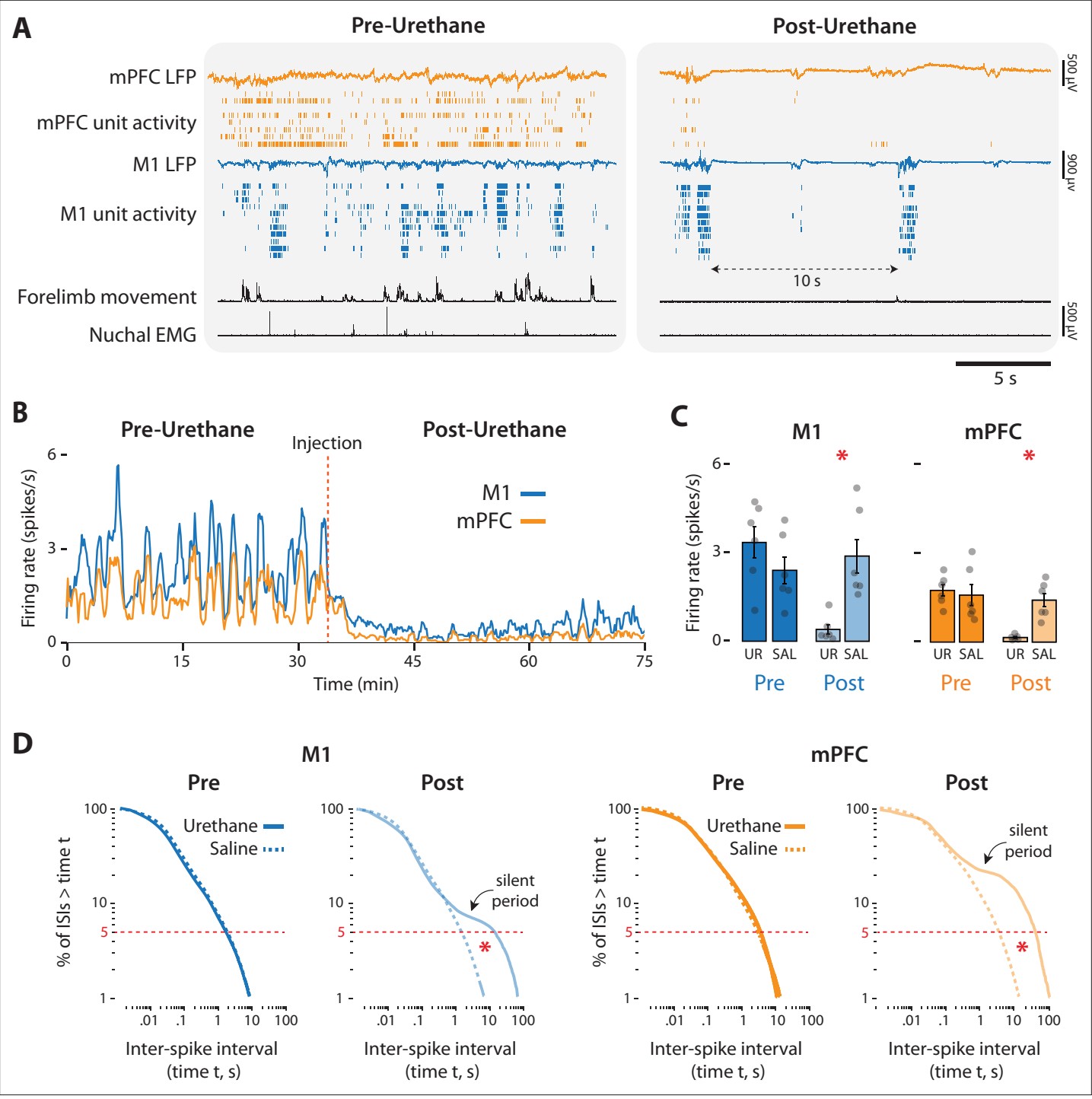

**Figure 7.** Urethane anesthesia suppresses unit activity in M1 and mPFC in P8 rats. (**A**) Representative 20-s segments of data from recordings in M1 and mPFC before (left) and after (right) injection of urethane (1.0 mg/g b.w.). For each record from the top, data are presented as follows: mPFC local field potential (LFP) (gold trace), mPFC unit activity (gold ticks), M1 LFP (blue trace), M1 unit activity (blue ticks), forelimb movement, and nuchal electromyography (EMG). (**B**) Representative 75-min segment of data showing mean unit firing rate (in 2-s bins) in M1 (blue) and mPFC (gold) before and after injection of urethane (vertical dashed line). (**C**) Bar graphs showing mean firing rates of neurons across pups in M1 (left) and mPFC (right) during the pre-injection (Pre) and post-injection (Post) periods for the urethane (UR) and saline (SAL) groups. Mean firing rates for individual pups are shown as gray circles. Means ± standard error of the mean (SEM). (**D**) Left: Survivor plots of pooled interspike intervals (ISIs) for M1 units during the pre- and post-injection periods for pups in the urethane (solid blue line) and saline (dashed blue line). Right: Same as at left but for mPFC during the pre-injection (dark gold) and post-injection (light gold) periods. Asterisks denote significant difference ($p \leq 0.025$) between urethane and saline groups for ISI values at the bottom fifth percentile (dashed horizontal lines).

*Figure 7 continued on next page*

*Figure 7 continued*

The online version of this article includes the following figure supplement(s) for figure 7:

**Figure supplement 1.** Urethane anesthesia reduces the rate of spindle bursts in M1 and mPFC in P8 rats.

Urethane administration also disrupted neural activity in M1 and mPFC (*Figure 7A, B*), causing reductions in firing rate of over 85% (*Figure 7C*). Mean reductions in firing rate were significant for both M1 ($t_{(10)} = 3.83$, p = 0.003, Cohen's $D$ = 2.21) and mPFC ($t_{(10)} = 5.01$, p < 0.001, Cohen's $D$ = 2.94). Urethane also dramatically and significantly reduced the mean rate of spindle bursts in the two areas ($t_{(10)}$s ≥ 3.18, ps ≤ 0.01, Cohen's $D$s ≥ 1.83) (*Figure 7—figure supplement 1*).

Urethane also changed the temporal patterning of neural activity (*Figure 7A*). In the absence of urethane, neural activity in both areas exhibited the discontinuous pattern characteristic of cortical activity at P8 (*Golshani et al., 2009*; *van der Bourg et al., 2017*; *Glanz et al., 2021*). In contrast, urethane injection produced a burst-suppression pattern that is characteristic of general anesthesia as well as coma, hypothermia, and neonatal trauma (*Grigg-Damberger et al., 1989*; *Steriade et al., 1994*; *Hellström-Westas et al., 2006*; *Shanker et al., 2021*). This pattern, comprising population bursts separated by periods of relative silence lasting 10 s or longer, is illustrated by survivor plots of interspike intervals (ISIs) (*Figure 7D*): In both areas, whereas the pre-injection ISI distributions for urethane and saline are indistinguishable, the post-injection distributions for the urethane group deviates substantially from the saline group, especially for longer ISIs where the pronounced shoulders in the plots, indicative of inter-burst silence, are evident. For the bottom fifth percentile of ISIs, we found a significant difference between urethane and saline groups during the post-injection period in both M1 and mPFC ($t_{(5)}$s ≥ 3.36; ps ≤ 0.007; Cohen's $D$s ≥ 1.94), but not during the pre-injection period ($t_{(5)}$s ≤ 1.64).

In summary, urethane anesthesia at P8 eradicates sleep–wake cycling, suppresses behavior, and produces atypical neural activity.

## Discussion

We demonstrate here in P8 and P12 rats that neurons in two prefrontal areas—M2 and mPFC—exhibit state-dependent neural activity and responsivity to somatosensory stimuli. First, at both ages, neural activity in M2 and mPFC increases specifically during AS, similar to previous findings at these ages in M1 and S1 (*Tiriac et al., 2014*; *Dooley et al., 2020*; *Glanz et al., 2021*). Second, we find that neurons in M2 and mPFC respond to reafference arising from twitches and wake movements, and exafference arising from manual stimulation, with the proportion of responsive neurons generally being highest in M1 and decreasing across M2 and mPFC. Finally, we show that urethane thwarts accurate assessments of brain–behavior relations in developing cortex by suppressing neural activity and abolishing sleep–wake states, thus explaining discrepancies between the present and previous findings. Altogether, these results highlight the potential importance of sleep and sensory experience for the functional development of prefrontal cortex.

### Prefrontal cortex is most active during sleep

In developing rats, AS modulates spiking and oscillatory activity in M1 and S1 (*Blumberg et al., 2020*; *Dooley et al., 2020*; *Glanz et al., 2021*), findings that we now extend to M2 and mPFC. Neural activity in these two areas was highest during movement-related periods of AS, but it was also higher during AS than wake even in the absence of movement. These findings suggest that state-dependent neuromodulation is a general feature of infant cortical activity. Although neuromodulators like acetylcholine and serotonin, respectively, influence early cortical activity (*Hanganu et al., 2007*; *Janiesch et al., 2011*) and development (*Kolk and Rakic, 2022*), it is not yet known whether these and other neuromodulators are released in a state-dependent fashion, as is known to occur in adults (*Lee and Dan, 2012*; *Jones, 2020*).

### Prefrontal cortex responds to sensory input

Sensory experience in early life scaffolds developing sensory and sensorimotor systems, providing information about the growing body and the world it inhabits (*Blumberg, 2015*). Notably, recent evidence from the visual system of both rodents and primates suggests that how sensory input reaches

cortex is fundamentally different in infants and adults. In adults, sensory information flows through a hierarchical network, from primary cortical areas to higher-order cortical areas. But, in the developing visual system, both primary and higher-order visual areas receive parallel sensory input directly from thalamus (*Warner et al., 2012*; *Murakami et al., 2022*). Likewise, in the developing sensorimotor system, both M1 and S1 receive parallel sensory input (*Dooley and Blumberg, 2018*; *Gómez et al., 2021*). This ascending sensory input to M1 and S1 may refine somatotopy and connectivity within and between these cortical areas, thus laying a foundation for their further development, including the later emergence of M1's motor functionality (*Dooley and Blumberg, 2018*; *Gómez et al., 2021*).

M2 is a higher-order sensorimotor area with a somatotopic representation, though its organization is coarser than M1's (*Mohammed and Jain, 2014*; *Mohammed and Jain, 2016*). Thus, it is perhaps not surprising that we found that M2 units, like those in M1, exhibit short-latency sensory responses to self- and other-generated forelimb movements. In contrast, mPFC does not exhibit somatotopic organization. Indeed, it has been theorized that the development of higher-order functions in prefrontal cortex derives in part from its relative independence from sensory input, relying instead on intrinsic (or spontaneous) neural activity (*Johnson et al., 2015*; *Johnson et al., 2021*; *Werchan and Amso, 2017*). Thus, it was surprising to find that mPFC units exhibit short-latency sensory responses with profiles that are nearly identical with those in M1 and M2 (*Figure 5B, C*).

The similar response profiles in M1, M2, and mPFC imply that the sensory events were triggered by the same subcortical source, as is the case with M1 and S1 at these ages (*Gómez et al., 2021*). However, our attempts to identify this source yielded null results. Thus, the source of M2 and mPFC's short-latency sensory responses in early development remains unknown, and likely differs from the source for M1 and S1. Moving forward, future studies may need to combine precise stimulation of ascending pathways with recordings in M2 and mPFC.

Across M1, M2, and mPFC, we observed a declining proportion of sensory-responsive units, with the highest proportion in M1 and the lowest in mPFC. Importantly, because mPFC is not somatotopically organized, by testing only forelimb sensory input it is likely that we are underestimating the proportion of mPFC units that respond to somatosensory input. Thus, the decreasing sensory responsiveness across M1, M2, and mPFC likely reflects their decreasing somatotopic homogeneity (*Figure 8*; *Asanuma and Mackel, 1989*; *Bedwell et al., 2014*; *Barthas and Kwan, 2017*).

Further, there is no reason to believe that sensory responses in the developing mPFC are exclusive to the somatosensory system. In adults, mPFC receives multimodal inputs (*Hoover and Vertes, 2007*; *Bedwell et al., 2014*) and is responsive to auditory stimuli arriving from the posterior medial thalamic nucleus (*Martin-Cortecero and Nuñez, 2016*). Accordingly, we expect neurons in developing mPFC to respond to somatosensory input from other parts of the body, as well as visual, auditory, olfactory, and gustatory input. Thus, whereas unimodal sensory input to primary sensory areas enables the development of somatotopic homogeneity, multimodal sensory input to prefrontal cortex may enable the development of functional heterogeneity.

## Urethane abolishes brain–behavior relations

Until now, developmental investigations of rat prefrontal cortex were conducted in urethanized pups (*Brockmann et al., 2011*; *Bitzenhofer et al., 2015*). Although urethane is known to alter neural activity and behavior (*Dyer and Rigdon, 1987*; *Simons et al., 1992*; *Sorrenti et al., 2021*), prior studies have discounted the significance of these affects. In fact, it has been claimed that urethane

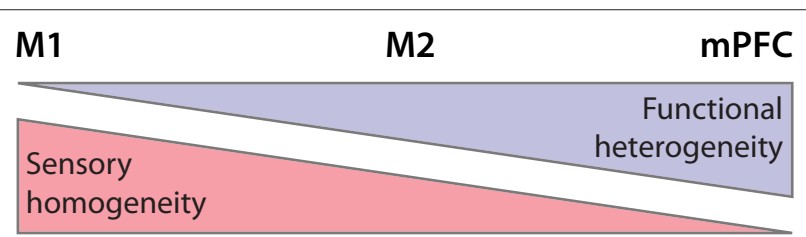

**Figure 8.** Summary illustration of sensory and functional gradients across M1, M2, and mPFC. Somatotopic organization (red) is more homogeneous in M1 than in M2 and mPFC. In contrast, functional organization (lavender) is more heterogeneous in mPFC than in M2 and M1.

mimics natural sleep in infant and adult rodents (*Clement et al., 2008*; *Pagliardini et al., 2013*; *Bitzenhofer et al., 2015*).

The present findings cannot be reconciled with these claims. Despite using a light dose in P8 rats, urethane eradicated natural sleep–wake states and profoundly suppressed neural activity and limb movements. Also, firing rates and oscillatory activity in M1 and mPFC exhibited a burst-suppression pattern that is not observed in healthy unanesthetized pups (*Grigg-Damberger et al., 1989*; *Steriade et al., 1994*; *Hellström-Westas et al., 2006*; *Iyer et al., 2014*; *Shanker et al., 2021*). Finally, we were unable to assess the effects of urethane on the sensitivity of mPFC to reafferent and exafferent stimuli because, respectively, urethane suppressed self-generated movements and exafferent processing was nearly absent even in unanesthetized pups.

Thus, as is increasingly appreciated in adults (*Akeju and Brown, 2017*; *Mondino et al., 2022*), anesthesia in infants is not a suitable proxy for natural sleep or sleep-related neural activity. Nor is it compatible with the goal of understanding brain–behavior relations in early development. Accordingly, extreme caution is warranted when interpreting results from anesthetized infant animals.

## Conclusions

Consistent with previous findings in sensorimotor cortex, we find that behavioral states and sensory experiences influence neural activity in developing PFC. These findings raise the possibility that the development of prefrontal cortex is influenced by more than intrinsic activity alone and that—similar to the discovery of the sensory foundations of 'motor cortex' (*Chakrabarty and Martin, 2005*; *Dooley and Blumberg, 2018*; *Glanz et al., 2021*; *Singleton et al., 2021*)—the early activity in PFC may not be related in obvious ways to its higher-order functions in adults. Thus, we propose that state-dependent modulation and sensory responsiveness are general features of developing cortex. In other words, the early functional development of primary and higher-order cortical areas may be more similar than currently appreciated.

Finally, although the similarities and differences between rodent and primate PFC have been debated for decades (*Preuss, 1995*; *Uylings et al., 2003*; *Barthas and Kwan, 2017*; *Carlén, 2017*; *Laubach et al., 2018*), there is still no consensus regarding the extent to which non-primates have cortical areas homologous to primate PFC. When defining PFC, different researchers variously emphasize anatomical connectivity and functional criteria, resulting in terminological confusion. For example, in rodents, M2 goes by many names and may perhaps be considered part of mPFC (*Barthas and Kwan, 2017*). The current study does not resolve these issues and was not designed to do so, but does suggest that a developmental-comparative approach will prove useful for clarifying PFC's evolutionary and functional history, as it has for other cortical domains (*Krubitzer and Dooley, 2013*).

# Materials and methods

**Key resources table**

| Reagent type (species) or resource | Designation | Source or reference | Identifiers | Additional information |
|---|---|---|---|---|
| Strain, strain background (*Rattus norvegicus*) | Sprague-Dawley Norway Rats | Envigo | | |
| Commercial assay or kit | Vybrant DiI Cell-Labeling Solution | Life Technologies | Cat #: V22885 | |
| Chemical compound, drug | 3,3'-Diaminobenzidine Tetrahydrochloride (DAB) | Spectrum | TCI-D0078-5G | |
| Chemical compound, drug | Isoflurane | Phoenix Pharmaceuticals | Item #: 0010250 | 3–5% |
| Chemical compound, drug | Ketamine hydrochloride | Akorn Animal Health | NDC: 59399-114-10 | 10:1 with xylazine (cocktail: >0.08 mg/kg, IP) |
| Chemical compound, drug | Xylazine | Lloyd Laboratories | sc-362949Rx | 1:10 with ketamine (cocktail: >0.08 mg/kg, IP) |
| Chemical compound, drug | Catalase from bovine liver | Sigma-Aldrich | C9322 | |
| Chemical compound, drug | Cytochrome c from equine heart | Sigma-Aldrich | C2506 | |
| Chemical compound, drug | Carprofen | Putney | #200-522 | 0.1 mg/kg |

*Continued on next page*

*Continued*

| Reagent type (species) or resource | Designation | Source or reference | Identifiers | Additional information |
|---|---|---|---|---|
| Chemical compound, drug | Bupivacaine | Pfizer | NDC 0409-1162-19 | 0.25% |
| Chemical compound, drug | Urethane | Sigma-Aldrich | CAS #: 51-79-6 | 1.0 mg/g b.w. |
| Software, algorithm | MATLAB, version 2020a | Mathworks | RRID: SCR_001622 | |
| Software, algorithm | Spike2, version 8 | Cambridge Electronic Design | RRID: SCR_000903 | |
| Software, algorithm | Adobe Illustrator | Adobe | RRID:SCR_010279 | |
| Software, algorithm | Adobe Photoshop | Adobe | RRID:SCR_014199 | |
| Software, algorithm | Spinview | FLIR | https://www.flir.com/products/spinnaker-sdk/ | |
| Software, algorithm | Kilosort | Marius Pachitariu | https://github.com/MouseLand/Kilosort; *Pachitariu, 2023* | |
| Software, algorithm | Phy2 | The Cortical Processing Laboratory at UCL | https://github.com/cortex-lab/phy; *Rossant and Harris, 2022* | |
| Software, algorithm | SPSS | IBM | RRID:SCR_019096 | |
| Software, algorithm | Synapse | Tucker Davis Technologies | https://www.tdt.com/component/synapse-software/ | |
| Other | Vetbond | 3M | https://www.3m.com/ | Tissue adhesive |

## Experimental animals

All experiments were conducted in accordance with the National Institutes of Health Guide for the Care and Use of Laboratory Animals (NIH Publication No. 80-23) and were approved by the Institutional Animal Care and Use Committee of the University of Iowa (protocol # 0021955).

Sprague-Dawley rats at P8–9 (hereafter 'P8'; body weight: 20.76 ± 1.77 g) and P12–P13 (hereafter 'P12'; body weight: 30.74 ± 2.16 g) were used. Pups were born to dams housed in standard laboratory cages (48 × 20 × 26 cm) with a 12-hr light/dark cycle. Food and water were available ad libitum. The day of birth was considered P0 and litters were culled to eight pups by P3. To protect against litter effects, pups selected from the same litter were always assigned to different experimental groups (*Abbey and Howard, 1973*; *Lazic and Essioux, 2013*). In addition, pups were randomly assigned to experimental groups.

## Experiment 1: Recordings in unanesthetized pups

In Experiment 1, we recorded M1, M2, and mPFC activity from unanesthetized rats at P8 and P12.

### Experimental procedure
### Surgical preparation

Surgery was performed using established methods (*Blumberg et al., 2015*; *Glanz et al., 2021*). Briefly, on the day of recording a pup of healthy weight and with a visible milk-band was removed from the litter, anesthetized with isoflurane (3.5–5%, Phoenix Pharmaceuticals, Burlingame, CA), and placed on a heating pad. Bipolar electrodes (California Fine Wire, Grover Beach, CA) were inserted into the nuchal muscle and left and right forelimb muscles (*biceps brachii*) and secured with collodion. An anti-inflammatory agent (Carprofen, 0.1 mg/kg SC; Putney, Portland, ME) was administered and the torso of the pup was wrapped in soft surgical tape. The scalp was sterilized with iodine and ethanol, and a portion of the scalp was removed to reveal the skull; a topical analgesic (bupivacaine, 0.25%; Pfizer, New Work, NY) was applied to the skull surface and surrounding skin, and then a veterinary adhesive (Vetbond; 3M, St. Paul, MN) was used to secure the skin to the skull. A steel head-fix (Neurotar, Helsinki, Finland) was attached to the skull using super glue (Loctite; Henkel Corporation,

Westlake, OH) and dried with accelerant (INSTA-SET; Bob Smith Industries, Atascadero, CA). The pup was secured in a stereotaxic apparatus (Kopf Instruments, Tujunga, CA) and, under isoflurane anesthesia, a steel trephine (1.8 mm; Fine Science Tools, Foster City, CA) was used to drill openings in the skull over forelimb M1 (all coordinates from bregma; P8: +1.0 mm rostrocaudal (RC), 1.8 mm mediolateral (ML); P12: +1.0 mm RC, 1.8–2.0 mm ML), forelimb M2 (P8 and 12: +2.0 mm RC, 1.0 mm ML), and mPFC (P8 and P12: +1.8 mm RC, 0.5 mm ML). The pup was then transported to the recording rig, where it recovered for at least 1 hr. Recording began only after regular sleep–wake cycles were observed and intracranial temperature reached 36°C.

## Data acquisition

Neurophysiological and EMG data were collected using a data acquisition system (Tucker-Davis Technologies, Gainsville, FL) with sampling rates of approximately 25 and 1.5 kHz, respectively. Neural data were collected simultaneously from two cortical locations using 16-site, 3 or 5 mm silicon-iridium electrodes (A1x16-3 mm-100-177-A16 or A1x16-5 mm-100-177-A16; NeuroNexus, Ann Arbor, MI). Before insertion, electrodes were coated with a fluorescent dye (DiI; Invitrogen, Waltham, MA) for later confirmation of placement. A chlorinated silver wire (0.25 mm in diameter; Medwire, Mt. Vernon, NY) was inserted into occipital cortex and used as both reference and ground. Neural data were recorded and visualized using Synapse software (Tucker-Davis Technologies). Video was collected using a BlackFly-S camera (100 fps) and SpinView software (FLIR Integrated Systems, Wilsonville, OR). To enable synchronization of the video and electrophysiological records, an LED was positioned within the camera frame and was programmed to flash once every 3 s (*Dooley et al., 2021*).

## Experimental design

We performed two sequential recordings in 10 pups at P8 (five female) and 14 pups at P12 (seven female). We first recorded activity from the forelimb regions of M1 and M2. Electrodes were inserted into the target sites and allowed to settle for at least 10 min. To confirm electrode placements in the forelimb region of both areas before recording began, the experimenter used a cotton-tipped dowel to move the contralateral forelimb while monitoring neural activity. If forelimb-related activity was not detected during the first electrode placement (a rare occurrence), the electrode was withdrawn, repositioned, and lowered again; electrodes were never repositioned more than twice. Video and neurophysiological data were recorded for 40 min as the pup cycled freely between sleep and wake. This period was followed by 50 manual stimulations of the right forelimb (as described above), delivered approximately 2–3 s apart. Upon completion of the stimulation protocol, the electrode in M2 was carefully withdrawn, coated again with DiI, and reinserted into mPFC (the M1 electrode was not disturbed). After the electrode settled in mPFC for at least 10 min, we again recorded video and neurophysiological data for 40 min, followed by 50 stimulations of the right forelimb. In total, the M1–M2 dataset consisted of eight recordings at P8 and nine at P12; the M1–mPFC dataset consisted of eight recordings at P8 and eight at P12.

## Data analysis
### Processing of neurophysiological data

Neurophysiological data were filtered for unit activity (bandpass: 300–5000 Hz) and converted into binary files. Templates for putative spikes were extracted using Kilosort (*Pachitariu et al., 2016*) and visualized using Phy2 (*Rossant and Harris, 2022*), as previously (*Dooley et al., 2021*; *Glanz et al., 2021*; *Gómez et al., 2021*). Spike waveforms and autocorrelations were used to identify single units and multiunits. Preliminary analyses were performed to confirm that the activity profiles of single units and multiunits did not differ in any systematic way. Thus, all subsequent analyses were conducted using both single-unit and multiunit activity (hereafter 'units' or 'unit activity'). To obtain local field potentials (LFPs), neurophysiological data were downsampled to ~1000 Hz, smoothed (0.005 s), and converted into binary files. Spike-time and LFP data were imported into MATLAB for analysis. To extract spindle bursts, LFP signals were bandpass filtered at 10–20 Hz (stopband attenuation: −−60 dB; transition

gap: 1 Hz) and the phase and amplitude of the filtered signal were calculated using a Hilbert transform (*Glanz et al., 2021*). Spindle bursts were defined as events for which the waveform amplitude exceeded, for at least 100 ms, the median amplitude plus two standard deviations of the baseline amplitude. Spindle-burst onset was determined using previously described methods (*Dooley et al., 2020*).

## Analysis of behavioral state

Motor activity and behavioral state were assessed visually using video and corroborated with EMG. We used custom-written MATLAB scripts to detect frame-by-frame changes in pixel intensity within user-defined regions of interest (*Dooley et al., 2021*). We selected two regions, one encompassing the right forelimb and the other encompassing the entire body, to allow detection of movement periods. Movements were represented as changes in pixel intensity across time. We then imported movement, neurophysiological, and EMG data into Spike2 (Cambridge Electronic Design, Cambridge, UK).

Recording data were separated into periods of AS and wake. AS was defined by the presence of myoclonic twitches occurring against a background of nuchal muscle atonia. Twitches appear as brief, jerky limb movements and as sharp spikes in the EMG record. Wake was defined by the presence of increased nuchal muscle tone relative to AS, most commonly initiated by and containing large-amplitude movements of multiple limbs (*Del Rio-Bermudez et al., 2020*; *Glanz et al., 2021*). Periods of behavioral quiescence that were not defined as AS or wake were examined but not included in the present analyses. Behavioral state was always scored blind to neural activity.

To quantify differences in firing rate across AS and wake, for each unit we calculated the mean firing rate over the duration of each state; we then calculated mean firing rate across units within each brain area for each pup. At P8, we also assessed whether the occurrence of spindle bursts was state dependent. For each area in each animal, we determined the mean rate of spindle bursts during AS and wake, and then calculated the mean rates across pups. Spectrograms of oscillatory activity were generated using the sonogram function in Spike2. (This analysis was not performed at P12 because spindle bursts are not clearly discernable at this age.)

To delineate differences between state- and movement-dependent changes in firing rate in M1, M2, and mPFC, we calculated the mean firing rate in each area during periods of movement. Movement and non-movement periods were extracted using custom MATLAB scripts from whole-body movement data (derived from video as described above). The onset of a movement period occurred when movement data exceeded a threshold value of 3× greater than baseline for at least 250 ms; the offset of a movement period occurred when movement data decreased below threshold. Movement and non-movement periods were categorized as to whether they occurred during AS or wake. Mean firing rates were calculated for each unit during the following conditions: AS with movement, AS with no movement, wake with movement, and wake with no movement. We determined the mean firing rate across units within an area and then the mean rate across pups.

## Analysis of sensory activity

We analyzed neural activity in M1, M2, and mPFC in response to twitches, wake movements, and stimulations (hereafter referred to collectively as 'sensory events'). First, we scored twitches of the right forelimb during AS. The onset of a twitch was defined as the first video frame showing movement. When a bout of rapid twitching was detected, the first twitch in the bout was always scored; subsequent twitches in the bout were also scored if they could be clearly distinguished from the previous twitch (e.g., by a change in movement direction). We also scored individual forelimb wake movements; because wake movements typically occur as bouts of long continuous sequences, only the first limb movement in a bout was scored. The onset time of each forelimb stimulation was scored as the first video frame in which the dowel touched the forelimb. Sensory events were also always scored blind to neural activity.

To determine whether units in M1, M2, and mPFC were responsive to sensory input, we constructed PETHs for unit activity triggered on sensory events. We calculated PETHs in spikes/s for each unit, triggered on sensory events (window = 1 s, offset = 0.5 s, bin size = 10 ms). We then defined a baseline window (BLW; −500 to −200 ms before the event) and a sensory-response window (RW; 0–200

ms after the event) and calculated the mean firing rate within each window. If the firing rate during the RW was greater than the mean baseline firing rate plus 2× the standard deviation of the baseline value ($\bar{x}_{RW} > \bar{x}_{BLW} + 2 * \sigma_{BLW}$), the unit was categorized as 'responsive'. Using only responsive units, we again calculated average PETHs within each structure and normalized the data to the maximum value within each PETH.

To assess the reliability of neural responses to sensory events, we examined the activation rate, defined as the percentage of events to which each area responded. Again, using only responsive units, we examined the activity of each unit before and after each sensory event, using the same BLW as above but increasing the RW to 0–300 ms after the event so that the window durations were equal. For a given unit within a cortical area, if the summed unit activity within the RW exceeded a threshold defined as 1.5× the summed unit activity within the BLW ($\sum RW > 1.5 * \sum BLW$), the unit was categorized as having an event response. When more than 30% of recorded units within a cortical area exceeded this threshold, it was determined that the area as a whole for that pup was activated (*Glanz et al., 2021*). If a pup had no responsive units in a given area, that area's activation rate was set to zero.

Finally, to assess whether spindle bursts are associated with sensory events in M1, M2, and mPFC, we calculated the probability that a spindle burst was preceded by a sensory event for each area. First, for each spindle burst, we defined a window of time spanning 500 ms before the spindle burst onset. If a sensory event occurred within that window, we inferred that the sensory event triggered the spindle burst. We then divided the number of spindle bursts preceded by a sensory event by the total number of spindle bursts to derive a probability for that behavioral state or stimulation period. Probabilities were calculated for each area in each animal, and then averaged across animals. To establish the probability that a spindle burst was preceded by a sensory event due to chance, we performed 100 random reshuffles of sensory-event and spindle-burst onset times within a behavioral state or stimulation period. We used the mean probability of these 100 reshuffles to derive the expected (chance) probability for each type of sensory event. We then compared the expected and observed probabilities to determine whether spindle bursts were reliably preceded by sensory events. Finally, we compared spindle-event probabilities across areas to determine if the reliability of a sensory event preceding a spindle was similar across areas.

## Experiment 2: Recordings in urethanized pups

In Experiment 2, we used P8 rats to assess the effects of urethane anesthesia on behavior and neural activity in M1 and mPFC.

### Experimental procedure
#### Surgical preparation

Surgical preparation. Pups were prepared for neurophysiological recording as in Experiment 1, with one exception: After securing EMG electrodes, a small (1-mm) incision was made in the skin near the base of the tail and surgical-grade silicon tubing (inner diameter: 0.020 in; outer diameter: 0.037 in; SAI Infusion Technologies, Lake Villa, IL) was secured subcutaneously with veterinary adhesive.

#### Data acquisition
Neurophysiological and video data were acquired as in Experiment 1.

#### Experimental design

We recorded from P8 rats before and after injection of urethane (*n* = 7) or saline (*n* = 6). We recorded baseline video and neurophysiological data for 30 min while pups cycled between sleep and wake, followed by 50 stimulations of the right forelimb as in Experiment 1. We then infused urethane (1.0 mg/g b.w.; Sigma-Aldrich, St. Louis, MO) or an equivalent volume of sterile saline (Fresenius Kabi, Bad Homburg, Germany) through the implanted cannula. This procedure minimized disruption of the pup and allowed for uninterrupted recording of data. Also, subcutaneous infusion of urethane produces a comparable level of surgical anesthesia as intraperitoneal injection and reduces the likelihood of organ puncture (*Maggi and Meli, 1986*; *Field and Lang, 1988*; *Matsuura and Downie,*

*2000*). We waited at least 10 min for the drug to take effect, after which data were again recorded for 30 min followed by 50 stimulations of the right forelimb. One pup was excluded from analysis due to the complete loss of neural activity after urethane administration.

## Data analysis
### Processing of neurophysiological and video data

Neurophysiological and video data were processed as in Experiment 1. However, because sleep–wake states were eradicated in the urethanized pups, they were not scored and analyzed.

### Analysis of changes in movement quantity

We compared the effects of urethane and saline administration on movement based on pixel-change data derived from video, as in Experiment 1. For each pup, we visualized the smoothed (0.01 s) whole-body movement data in Spike2 to determine the baseline level of activity. Baseline movement activity was based on the mean pixel change across five 1-s windows during which no movement occurred. Then, in MATLAB, we summed the movement data (in pixel intensity changes per frame) separately during the pre- and post-injection periods; for each period, we performed a baseline subtraction. The sums of the movement data in the pre- and post-injection periods were divided by the total duration of each period. Next, for each pup we calculated the percentage change in movement between the pre- and post-injection periods. Finally, we calculated the mean percentage change across pups for each experimental group.

### Analysis of changes in firing rate

We calculated the mean firing rate of each unit in M1 and mPFC during the pre- and post-injection periods. For each pup, we calculated mean unit firing rate within each area, and then calculated mean firing rates for each area across pups in the two experimental groups.

### Analysis of ISIs
For each pup, we calculated the bottom fifth percentile of ISIs during the pre- and post-injection periods for the two experimental groups.

### Analysis of changes in spindle-burst activity

We identified spindle bursts from LFP data as described in Experiment 1, using the pre-injection period to calculate the baseline LFP amplitude. The rate of spindle bursts was calculated for the pre- and post-injection periods.

## Histology
At the end of each experiment, pups were overdosed with ketamine–xylazine (>0.08 mg/kg, intraperitoneal) and perfused transcardially with phosphate-buffered saline (PBS, 1 M) followed by 4% paraformaldehyde (PFA). The brain was extracted and fixed for at least 24 hr in PFA and 48 hr in phosphate-buffered sucrose. Brain tissue was sliced coronally (80 µm) using a freezing microtome (Leica Biosystems, Wetlzar, Germany) and tissue was wet-mounted to locate the electrode tracks using fluorescence microscopy (×2.5–5 magnification; Leica Microsystems). Sections were placed in well plates and stained for cytochrome oxidase (CO) to visualize cortical layers. Well plates were filled with CO solution (catalase, cytochrome C, DAB, phosphate-buffered $H_2O$, and $DiH_2O$) and sections were allowed to develop in solution for 3–6 hr on a heating pad (*Dooley et al., 2021*). Sections were then rinsed with PBS, slide-mounted, and allowed to dry for 48 hr; slides were then placed in citrus clearing solution for 5 min, after which they were cover-slipped with dibutylphthalate polystyrene xylene (DPX) mounting medium. Cover-slipped slides were allowed to dry for at least 24 hr. Fluorescent and brightfield images (at ×2.5–5 magnification) were imported into Adobe Illustrator (San Jose, CA) and electrode tracks were reconstructed. CO-stained slides were used to determine the border

between S1 and M1 (using layer 4 as a boundary). We demarcated mPFC and M2 based on the structure and orientation of cortical layers and with the aid of brain atlases (*Paxinos and Watson, 2009*; *Khazipov et al., 2015*).

## Statistical analyses

All statistical analyses were performed in SPSS (IBM) and MATLAB. Probabilities and percentages were arcsine transformed before statistical testing to correct for edge effects. For all tests, $\alpha$ was set to 0.05, unless otherwise specified; when appropriate, the Bonferroni correction procedure was used. A Shapiro–Wilk test was used to assess normality. We tested for significance using repeated-measures ANOVA and paired and unpaired $t$ tests. Means are reported with their standard error (SEM). We report effect sizes for ANOVA as partial eta square ($\eta_p^2$) and for $t$ tests as Cohen's $D$.

## Acknowledgements

This research was supported by a grant from the National Institute of Child Health and Human Development (R37-HD081168) to MSB. We thank Greta Sokoloff, Ryan Glanz, and Toby Mordkoff for helpful comments and advice.

## Additional information

### Funding

| Funder | Grant reference number | Author |
|---|---|---|
| National Institutes of Health | R37-HD081168 | Mark S Blumberg |

The funders had no role in study design, data collection, and interpretation, or the decision to submit the work for publication.

### Author contributions

Lex J Gómez, Conceptualization, Data curation, Software, Formal analysis, Investigation, Visualization, Methodology, Writing – original draft, Writing – review and editing; James C Dooley, Conceptualization, Supervision, Visualization, Methodology, Writing – review and editing; Mark S Blumberg, Conceptualization, Resources, Supervision, Visualization, Methodology, Project administration, Writing – review and editing

### Author ORCIDs

Lex J Gómez ⬤ http://orcid.org/0000-0003-4643-0869
James C Dooley ⬤ http://orcid.org/0000-0002-9868-9840
Mark S Blumberg ⬤ http://orcid.org/0000-0001-6969-2955

### Ethics

All experiments were conducted in accordance with the National Institutes of Health Guide for the Care and Use of Laboratory Animals (NIH Publication No. 80-23) and were approved by the Institutional Animal Care and Use Committee of the University of Iowa. All surgery was performed under isoflurane anesthesia, and every effort was made to minimize suffering.

### Decision letter and Author response

Decision letter https://doi.org/10.7554/eLife.82103.sa1
Author response https://doi.org/10.7554/eLife.82103.sa2

## Additional files

### Supplementary files
• MDAR checklist

## Data availability

Whenever possible, individual data points are represented within a figure. Raw data (including timestamps of action potentials, oscillatory events, behavioral events, and behavioral states) have been uploaded to Dryad (https://doi.org/10.5061/dryad.18931zd1w). Select custom MATLAB scripts are available on Github (https://github.com/lexjgomez/Gomez_et_al_2023, (copy archived at swh:1:rev:53d289b76363d841590cb78182402175c997887d)).

The following dataset was generated:

| Author(s) | Year | Dataset title | Dataset URL | Database and Identifier |
|---|---|---|---|---|
| Gomez L, Dooley J, Blumberg M | 2022 | Activity in developing prefrontal cortex is shaped by sleep and sensory experience | https://dx.doi.org/10.5061/dryad.18931zd1w | Dryad Digital Repository, 10.5061/dryad.18931zd1w |

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
