## [Editor Report]

This manuscript examines neural activity in several cortical areas (such as the primary and secondary motor cortex and the medial prefrontal cortex) across sleep-wake states and under anesthesia. The quality of the recordings in infant rats is excellent, evidence is solid, and results are important in the field of research into the role of active sleep in the neuronal and circuit mechanisms of early cortical development. Some of the findings presented and the hypothesis developed are novel, and some should hopefully prompt future developmental studies to look at sleep as an essential component that cannot be replaced by using anesthetics.

---

## [Decision Letter]

**Decision letter after peer review:**

Thank you for submitting your article "Activity in developing prefrontal cortex is shaped by sleep and sensory experience" for consideration by *eLife*. Your article has been reviewed by 2 peer reviewers, and the evaluation has been overseen by a Reviewing Editor and Laura Colgin as the Senior Editor. The following individuals involved in review of your submission have agreed to reveal their identity: Lukas I Schmitt (Reviewer #1); Anita Lüthi (Reviewer #2).

Essential revisions:

(1) Further analysis: Data presented are mainly descriptive and remain at an observational state rather than bringing functional insights. Further analysis are absolutely required on, for example: (1) the nature of the relationship of single unit firing with sleep states, spindle activity, sensory inputs or motor activity across different development stages, (2) correlation between single unit responses with the different types of events investigated, spike train variability, unit properties description, correlation between M1-M2 units….

(2) Further analysis must also include more statistics.

(3) The authors should consider toning down some of their claims that are not sufficiently substantiated by the actual experiments. For instance, the effects of sensory stimulation under urethane anesthesia have not been performed but could be useful to argue on the effects of urethane. If the authors choose to run new experiments, they can publish those later as a Research Advance that will be associated with this paper, should it ultimately be accepted. Alternatively, should the authors already have data available that suggests possible neural substrates of early sensory responsiveness (e.g., histological data regarding the connectivity of motor and prefrontal areas at these early stages of development), then you may want to consider adding these additional data.

(4) The term sensory feedback seems to be inappropriately used.

*Reviewer #1 (Recommendations for the authors):*

The manuscript should be revised with additional analysis included. Additional experiments may also be needed if some analysis cannot be performed on the data already collected. Suggestions for these additional analysis and potential experiments as well as critical issues with presentation are listed below.

(1) As currently presented, analysis of the recordings made by the authors are largely descriptive and so their findings are difficult to interpret beyond the initial observation of state-related activity in the three areas examined. While it is true, as the authors state that "These findings expand the range of factors shaping the activity-dependent development of higher-order cortical areas", it is not particularly clear what exactly these factors are or how the activity observed may be involved in development. To address this, the authors should further analyze recorded activity to clarify the nature of the relationship of single unit firing with active sleep as well as with spindle activity, sensory input, and movements across the developmental ages examined. For example, to link activity at the single unit level with spindles, the authors could assess spindle-associated firing rates and spike phase relationships in recorded units across areas (i.e. as in Dickey et al., Nat. Comm. 2021). PETH histograms like those used for motor and sensory events could also be useful for this purpose. More generally, assessment of the correlation between responses of single units to different types of events (i.e. in figure 5), at least for responding neurons, would help to determine whether there are connections between activity elicited.

(2) Analysis of the effects of Urethane, while informative, is incomplete in that the authors do not connect the observed effects of Urethane administration with their other observations. For instance, it does not appear that the response to sensory events is assessed. If there is another means to more directly link the observed urethane dependent changes with their other analysis, then this experiment (sensory stimulation under urethane anesthesia) should be performed. It would also be important to compare effect size across areas to support the idea that urethane obscures state-related activity specifically in higher cortical areas such as PFC.

(3) In the discussion, the authors consider differences across the areas recorded and suggest that the "decreasing somatotopic homogeneity and increasing functional heterogeneity" of PFC relative to M1 suggests that "sensory input in early development likely serves different functional roles" for these areas. While this idea is interesting and the illustration of this claim in Figure 8 suggests that the authors think it is important for their overall message, the supporting evidence included in the manuscript is relatively weak. It is not clear that the observation of different percentages of responsive neurons in different areas shows any sort of functional heterogeneity and, in any case, it is not clear what this would mean from a functional perspective. If the authors want to include this claim, which I think does add to the value of the study overall, they should include additional analysis such as correlating single unit responses to different types of events. Comparison of spike-train variability (e.g. Ruyter vavn Steveninck et al. science 1997) across areas may also be useful for this purpose.

*Reviewer #2 (Recommendations for the authors):*

The study is very well done and the data is clearly described. One wonders whether the authors could go a step further to provide more analytic insight. Do these areas receive synaptic projections from subcortical somatosensory areas? Are these innervation patterns consistent with the decline of responsiveness from P8 to P12? And what is the role of M1-M2 or M1-mPFC projections?

The study could also go a step further in the analysis of the existing data. For example, a presentation of unit properties is missing, which leaves the reader uninformed about unit heterogeneity and distribution across layers.

The non-expert reader may wonder about the general description of twitches, wake-related movements and external sensory stimuli as "sensory feedback". It is not clear what is meant with feedback during external sensory stimuli. It is also unclear what role proprioceptive stimuli, e.g. the ones from muscle spindles play here. This also raises the question could look at whether individual units share responsiveness to twitches, wake movements and stimulation.

Figure 1C intriguingly suggests that M2 units discharge in a manner that seems to be complementary to the one observed for M1 units. This seems much less to be the case for mPFC units. This would deserve some more in-depth analysis. This in particular is because the mean firing plots in Figure 2 do not seem to replicate this. How is the dataset shown in Figure 1 related to the one in Figure 2?

Even though the criteria to score active sleep and to distinguish it from wakefulness seem clear to the authors, this is not obvious in the short data stretches presented. In particular, the authors description of a nuchal atonia is not evident in Figure 1C. What were the thresholds set for this atonia?

It would also be great to have some more quantitative information about the distinct LFP activity patterns in M2 and PFC at P8 and P12. Why were they not taken into account to score states of vigilance? What are the frequencies of the rhythmic activity pattern evident at P12 but not at P8 that seems typical for active sleep? Time-frequency plots of the kinds shown in Figure 3 would be helpful to get a more quantitative insight.

I am not sure the time-frequency plot shown in Figure 3A is really helpful. Here, instead of such a plot, a dataset illustrating the presence of spindle bursts in unfiltered and band-pass filtered data would instead be helpful to recognize the individual events.

Does the LED light that was used for synchronization purposes disrupt the sleep-wake activity of these pups?

Line 133-134. This sentence is difficult to read: what are periods of AS-related periods of quiescence?

Line 423 specify what the 5 ms time indications is for the smoothing – a sliding window length or a filter?

438 behavioral states, what is the distribution

Figure 1B should be shown at higher magnification

[Editors' note: further revisions were suggested prior to acceptance, as described below.]

Thank you for resubmitting your work entitled "Activity in developing prefrontal cortex is shaped by sleep and sensory experience" for further consideration by *eLife*. Your revised article has been evaluated by Laura Colgin (Senior Editor) and a Reviewing Editor.

The manuscript has been improved but there are some remaining issues that need to be addressed, as outlined below:

*Reviewer #2 (Recommendations for the authors):*

In my view, the authors have improved the manuscript and addressed some important issues with the original submission. In particular, the assessment of event-associated response characteristics in Figure 3, S1 is helpful for understanding spindle-associated activity. Nonetheless, some further revisions and clarifications would be important prior to acceptance. These suggestions are detailed below:

(1) One issue pointed out in the review of the first submission was the apparent disconnect between the investigation of sleep-associated regional dynamics that are the major focus of the study and the latter section examining the effects of Urethane. The authors' explanation as to the reason for not including an assessment of mPFC responses under anaesthesia presented in their response to review is reasonable but this point should be made more clearly in the discussion.

(2) Although the authors now include a discussion regarding links between M1, M2, and PFC, relevant to the dynamics they observe, this still does not really address the question of whether processing across these regions occurs in parallel or as part of a connected series. While it is difficult to address this point conclusively, as mentioned in the manuscript the similarity in the response profile peak timing across regions suggests that they are engaged in parallel (e.g. in Figure 5). To make this point more formally, it might be helpful to show the distribution of peak latency from threshold across the responding units in each region for the different conditions and developmental stages. This would help to more completely address the concerns raised in "Essential Revisions 3". Alternatively, the authors could present the evidence against the coactivation of neurons across the recorded regions mentioned in line 185 but not currently included in the manuscript.

*Reviewer #3 (Recommendations for the authors):*

The authors have responded to my comments and have improved the manuscript by:

- Clarifying their scoring procedure;

- Adding additional datasets on unit activity in two new figures;

- Changing an existing figure by adding more representative data.

I also like a lot the newly added discussion part that makes the significance of the present findings more clear. In fact, that PFC is responsive to sensory stimuli at early stages of development is a game changer in how one should think about cortical development. Furthermore, the comparison between anesthesia and sleep is also helpful. It should hopefully prompt future developmental studies to look at sleep as an essential component that cannot be replaced by using anesthetics.

---

## [Author Response]

Essential revisions:(1) Further analysis: Data presented are mainly descriptive and remain at an observational state rather than bringing functional insights. Further analysis are absolutely required on, for example: (1) the nature of the relationship of single unit firing with sleep states, spindle activity, sensory inputs or motor activity across different development stages, (2) correlation between single unit responses with the different types of events investigated, spike train variability, unit properties description, correlation between M1-M2 units….

As suggested, we added unit analyses of state-dependent activity from a randomly selected animal at each age (new Figure 2, Supplement 1). This figure shows that individual units are modulated by behavioral state. We also show a relationship between unit activity and spindle activity (new Figure 3, Supplement 1).

(2) Further analysis must also include more statistics.

We now include more statistical analyses as appropriate throughout the paper. For example, we performed analyses on the inter-spike interval data in Figure 7.

(3) The authors should consider toning down some of their claims that are not sufficiently substantiated by the actual experiments. For instance, the effects of sensory stimulation under urethane anesthesia have not been performed but could be useful to argue on the effects of urethane. If the authors choose to run new experiments, they can publish those later as a Research Advance that will be associated with this paper, should it ultimately be accepted. Alternatively, should the authors already have data available that suggests possible neural substrates of early sensory responsiveness (e.g., histological data regarding the connectivity of motor and prefrontal areas at these early stages of development), then you may want to consider adding these additional data.

We do have data showing prefrontal responses to exafferent stimulations under anesthetized vs. non-anesthetized conditions. However, as we state in our response to Reviewer #1, because of the low responsiveness of mPFC to exafferent stimulation, we are not able to draw definitive conclusions.

As to the issue of whether M1 is conveying sensory information to M2 and mPFC, this was our expectation; however, our findings do not support this idea, and we have added key sentences in the Discussion (lines 289 – 292) to address it.

(4) The term sensory feedback seems to be inappropriately used.

Thank you for catching this error! It has been corrected.

Reviewer #1 (Recommendations for the authors):The manuscript should be revised with additional analysis included. Additional experiments may also be needed if some analysis cannot be performed on the data already collected. Suggestions for these additional analysis and potential experiments as well as critical issues with presentation are listed below.(1) As currently presented, analysis of the recordings made by the authors are largely descriptive and so their findings are difficult to interpret beyond the initial observation of state-related activity in the three areas examined. While it is true, as the authors state that "These findings expand the range of factors shaping the activity-dependent development of higher-order cortical areas", it is not particularly clear what exactly these factors are or how the activity observed may be involved in development. To address this, the authors should further analyze recorded activity to clarify the nature of the relationship of single unit firing with active sleep as well as with spindle activity, sensory input, and movements across the developmental ages examined. For example, to link activity at the single unit level with spindles, the authors could assess spindle-associated firing rates and spike phase relationships in recorded units across areas (i.e. as in Dickey et al., Nat. Comm. 2021). PETH histograms like those used for motor and sensory events could also be useful for this purpose. More generally, assessment of the correlation between responses of single units to different types of events (i.e. in figure 5), at least for responding neurons, would help to determine whether there are connections between activity elicited.

We have modified the text to reflect how these findings expand the range of potential factors shaping activity-dependent development of higher-order cortical areas. To truly determine whether factors such as sleep and sensory input shape prefrontal development, future studies should manipulate these factors.

We have also added new supplementary figures addressing the activity of individual units during different behavioral states (Figure 2, Supplement 1) and the relationship between unit activity and spindle activity (Figure 3, Supplement 1).

In response to the reviewer’s comment, we used a variety of data visualization methods in an attempt to identify patterns in the sensory responsiveness of individual units. However, our experiments were not designed to address this question, and we were unable to draw meaningful conclusions from the data.

(2) Analysis of the effects of Urethane, while informative, is incomplete in that the authors do not connect the observed effects of Urethane administration with their other observations. For instance, it does not appear that the response to sensory events is assessed. If there is another means to more directly link the observed urethane dependent changes with their other analysis, then this experiment (sensory stimulation under urethane anesthesia) should be performed. It would also be important to compare effect size across areas to support the idea that urethane obscures state-related activity specifically in higher cortical areas such as PFC.

The primary aim of the urethane experiment was to demonstrate the disruptive effects of this anesthetic on mPFC activity and brain-behavior relations in P12 rats. That is indeed what we found.

Without the expression of sleep and wake behaviors under urethane, we could not assess reafferent responses in mPFC. And although we collected data regarding exafferent responses in mPFC to forelimb stimulation before and after urethane injection, the low responsiveness of mPFC units to forelimb stimulation in general (see Figure 5) made it impossible to draw meaningful conclusions.

(3) In the discussion, the authors consider differences across the areas recorded and suggest that the "decreasing somatotopic homogeneity and increasing functional heterogeneity" of PFC relative to M1 suggests that "sensory input in early development likely serves different functional roles" for these areas. While this idea is interesting and the illustration of this claim in Figure 8 suggests that the authors think it is important for their overall message, the supporting evidence included in the manuscript is relatively weak. It is not clear that the observation of different percentages of responsive neurons in different areas shows any sort of functional heterogeneity and, in any case, it is not clear what this would mean from a functional perspective. If the authors want to include this claim, which I think does add to the value of the study overall, they should include additional analysis such as correlating single unit responses to different types of events. Comparison of spike-train variability (e.g. Ruyter vavn Steveninck et al. science 1997) across areas may also be useful for this purpose.

We believe the reviewer misunderstood our argument. We revised this section of the Discussion to provide additional clarity. As reflected in the updated text, “functional heterogeneity” refers to the fact that whereas mPFC receives multimodal input, M1 and M2 primarily receive sensorimotor input.

As stated previously, individual unit analysis of responsiveness to different sensory events was inconclusive.

Reviewer #2 (Recommendations for the authors):The study is very well done and the data is clearly described. One wonders whether the authors could go a step further to provide more analytic insight. Do these areas receive synaptic projections from subcortical somatosensory areas? Are these innervation patterns consistent with the decline of responsiveness from P8 to P12? And what is the role of M1-M2 or M1-mPFC projections?

We do not have anatomical data regarding connectivity between thalamus and prefrontal cortex at these ages. However, evidence from the developing visual system indicates that sensory information is conveyed from the thalamus in parallel to both primary sensory areas and higher-order areas. As development progresses, the flow of sensory information transitions from parallel to serial (from primary sensory areas to higher-order areas). We have included this point and relevant citations in the Discussion.

The study could also go a step further in the analysis of the existing data. For example, a presentation of unit properties is missing, which leaves the reader uninformed about unit heterogeneity and distribution across layers.

We have added two new supplemental figures showing unit activity in all three cortical areas. The data in Figure 2, Supplement 1 show that individual units (from a single pup) in all three areas reliably mirror the state-dependent activity observed at the population level. We also show in Figure 3, Supplement 1 the relationship between unit activity and spindle activity.

Layer-specific activity could not be assessed in this study due to mPFC’s ventral midline location and thus the inaccessibility of its layers for laminar recordings.

The non-expert reader may wonder about the general description of twitches, wake-related movements and external sensory stimuli as "sensory feedback". It is not clear what is meant with feedback during external sensory stimuli. It is also unclear what role proprioceptive stimuli, e.g. the ones from muscle spindles play here. This also raises the question could look at whether individual units share responsiveness to twitches, wake movements and stimulation.

Our use of “sensory feedback” for exafferent input was an error and has been corrected.

As state previously, we were unable to draw meaningful conclusions from analyses of sensory responsiveness of individual units.

Figure 1C intriguingly suggests that M2 units discharge in a manner that seems to be complementary to the one observed for M1 units. This seems much less to be the case for mPFC units. This would deserve some more in-depth analysis. This in particular is because the mean firing plots in Figure 2 do not seem to replicate this. How is the dataset shown in Figure 1 related to the one in Figure 2?

We thank the reviewer for their careful attention to this figure. The data we presented in Figure 1C were not truly representative of the relations between M1 and M2 activity. We reviewed all our records and identified more representative data to include in this figure.

Even though the criteria to score active sleep and to distinguish it from wakefulness seem clear to the authors, this is not obvious in the short data stretches presented. In particular, the authors description of a nuchal atonia is not evident in Figure 1C. What were the thresholds set for this atonia?

We have updated this section of our Methods with additional detail on state scoring. We have updated the representative data in this figure to ensure that changes in nuchal tone are obvious at the scale shown in the figure.

It would also be great to have some more quantitative information about the distinct LFP activity patterns in M2 and PFC at P8 and P12. Why were they not taken into account to score states of vigilance? What are the frequencies of the rhythmic activity pattern evident at P12 but not at P8 that seems typical for active sleep? Time-frequency plots of the kinds shown in Figure 3 would be helpful to get a more quantitative insight.

At P8, state-typical oscillatory activity (e.g., δ) is not present and so cannot be used to identify behavioral state. Regardless, δ oscillations are only useful for assessing slow-wave sleep, which was not a focus of this paper. Our method of focusing on nuchal muscle tone and movements was sufficient for our study, as described in previous publications (Dooley and Blumberg 2018, eLife; Dooley et al. 2020, Current Biology; Glanz et al. 2021, Journal of Neuroscience; Gomez et al. 2021, Journal of Neuroscience).

I am not sure the time-frequency plot shown in Figure 3A is really helpful. Here, instead of such a plot, a dataset illustrating the presence of spindle bursts in unfiltered and band-pass filtered data would instead be helpful to recognize the individual events.

We present the data as shown to illustrate how overall oscillatory activity (including spindle bursts) differs across behavioral states.

Does the LED light that was used for synchronization purposes disrupt the sleep-wake activity of these pups?

The LED is not disruptive as the pups are tested in a dimly lit room (sleep-wake cycles are not dependent on the light cycles at these ages), their eyes are sealed shut, and the LED is out of view.

Line 133-134. This sentence is difficult to read: what are periods of AS-related periods of quiescence?

We meant periods of active sleep during which there was no movement. We have clarified the text.

Line 423 specify what the 5 ms time indications is for the smoothing – a sliding window length or a filter?

This is a moving filter window, as dictated by the MATLAB function “smooth.”

438 behavioral states, what is the distribution

The average time spent in each behavioral state is described in the Results.

Figure 1B should be shown at higher magnification

We have increased the magnification of 1B.

[Editors' note: further revisions were suggested prior to acceptance, as described below.]

Reviewer #2 (Recommendations for the authors):In my view, the authors have improved the manuscript and addressed some important issues with the original submission. In particular, the assessment of event-associated response characteristics in Figure 3, S1 is helpful for understanding spindle-associated activity. Nonetheless, some further revisions and clarifications would be important prior to acceptance. These suggestions are detailed below:(1) One issue pointed out in the review of the first submission was the apparent disconnect between the investigation of sleep-associated regional dynamics that are the major focus of the study and the latter section examining the effects of Urethane. The authors' explanation as to the reason for not including an assessment of mPFC responses under anaesthesia presented in their response to review is reasonable but this point should be made more clearly in the discussion.

We have addressed this comment by adding a sentence to the relevant section in the discussion (lines: 336–339).

(2) Although the authors now include a discussion regarding links between M1, M2, and PFC, relevant to the dynamics they observe, this still does not really address the question of whether processing across these regions occurs in parallel or as part of a connected series. While it is difficult to address this point conclusively, as mentioned in the manuscript the similarity in the response profile peak timing across regions suggests that they are engaged in parallel (e.g. in Figure 5). To make this point more formally, it might be helpful to show the distribution of peak latency from threshold across the responding units in each region for the different conditions and developmental stages. This would help to more completely address the concerns raised in "Essential Revisions 3". Alternatively, the authors could present the evidence against the coactivation of neurons across the recorded regions mentioned in line 185 but not currently included in the manuscript.

In previous studies, we used analytical methods, such as a shift predictor (Dooley and Blumberg 2018, Gómez et al. 2021), to determine whether sensory input in S1 and M1 is conveyed serially or in parallel from subcortical areas. That analysis yielded null results for the present dataset. Also, in response to the Reviewer’s comment, we evaluated the response latencies across sensory events, ages, and areas; unfortunately, this effort also did not yield interpretable results due in part to the variability of the individual data. However, to honor the Reviewer’s concern and add more clarity, we have modified language in the Results (lines: 195–203) and Discussion (lines: 300–306).